# OpenNucleome for high-resolution nuclear structural and dynamical modeling

**Zhuohan Lao, Kartik D Kamat, Zhongling Jiang, Bin Zhang\***

Department of Chemistry, Massachusetts Institute of Technology, Cambridge, United States

**Abstract** The intricate structural organization of the human nucleus is fundamental to cellular function and gene regulation. Recent advancements in experimental techniques, including high-throughput sequencing and microscopy, have provided valuable insights into nuclear organization. Computational modeling has played significant roles in interpreting experimental observations by reconstructing high-resolution structural ensembles and uncovering organization principles. However, the absence of standardized modeling tools poses challenges for furthering nuclear investigations. We present OpenNucleome—an open-source software designed for conducting GPU-accelerated molecular dynamics simulations of the human nucleus. OpenNucleome offers particle-based representations of chromosomes at a resolution of 100 KB, encompassing nuclear lamina, nucleoli, and speckles. This software furnishes highly accurate structural models of nuclear architecture, affording the means for dynamic simulations of condensate formation, fusion, and exploration of non-equilibrium effects. We applied OpenNucleome to uncover the mechanisms driving the emergence of 'fixed points' within the nucleus—signifying genomic loci robustly anchored in proximity to specific nuclear bodies for functional purposes. This anchoring remains resilient even amidst significant fluctuations in chromosome radial positions and nuclear shapes within individual cells. Our findings lend support to a nuclear zoning model that elucidates genome functionality. We anticipate OpenNucleome to serve as a valuable tool for nuclear investigations, streamlining mechanistic explorations and enhancing the interpretation of experimental observations.

**\*For correspondence:**
binz@mit.edu

## eLife assessment

This **important** work significantly advances the field of computational modeling of genome organization through the development of OpenNucleome. The evidence supporting the tool's effectiveness is **compelling** as the authors compare their predictions with experimental data. It is anticipated that OpenNucleome will attract significant interest from the biophysics and genomics communities.

## Introduction

The highly complex structural organization of the human nucleus plays a crucial role in the functioning and regulation of our cells (*Dekker et al., 2017*; *Hübner et al., 2013*; *Bickmore, 2013*; *Gorkin et al., 2014*; *Dekker and Mirny, 2016*; *Furlong and Levine, 2018*; *Finn and Misteli, 2019*; *Chen and Belmont, 2019*; *Lin et al., 2021*; *Liu et al., 2024*). The complexity arises from the diverse range of nuclear landmarks, such as nucleoli (*Lafontaine et al., 2021*), nuclear speckles (*Chen and Belmont, 2019*; *Lamond and Spector, 2003*), and the nuclear lamina (*van Steensel and Belmont, 2017*), each serving distinct functions. These landmarks provide specialized environments for various nuclear

processes, allowing for efficient coordination and regulation of gene expression. Moreover, the spatial arrangement of chromosomes within the nucleus, intertwined with the nuclear landmarks, is critical for proper gene regulation and communication between different genome regions. Disruptions or abnormalities in the nuclear organization can have profound consequences on cellular function and can contribute to the development of diseases, including cancer and genetic disorders (*Seruga et al., 2008*; *Schuster-Böckler and Lehner, 2012*).

Recent advancements in experimental techniques have significantly enhanced our understanding of nuclear organization (*Bickmore, 2013*; *Schmitt et al., 2016*; *McCord et al., 2020*; *Parmar et al., 2019*; *Jerkovic and Cavalli, 2021*; *Chen et al., 2016*). The advent of high-throughput sequencing-based methods, such as genome-wide chromosome-conformation capture (Hi-C), has unveiled crucial structural elements of the genome (*Dekker et al., 2002*; *Lieberman-Aiden et al., 2009*), including chromatin loops (*Rao et al., 2014*), topologically associating domains (*Dixon et al., 2016*; *Dekker and Heard, 2015*), and compartments (*Lieberman-Aiden et al., 2009*). Additionally, sequencing-based techniques such as DamID (*Greil et al., 2006*), Chip-Seq (*Park, 2009*), and TSA-Seq (*Chen et al., 2018*) have revealed valuable information regarding interactions between chromosomes and nuclear landmarks. However, it is worth noting that these sequencing methods often offer averaged contacts, which can mask the heterogeneity present across populations, although single-cell techniques are also emerging (*Wen et al., 2020*; *Ramani et al., 2017*; *Nagano et al., 2013*). Moreover, translating contact data into spatial positions can be challenging, adding complexity to interpreting experimental findings.

To complement these sequencing approaches, microscopic imaging techniques directly probe the spatial positions within individual nuclei (*Bickmore, 2013*; *van Steensel and Belmont, 2017*; *Chen et al., 2015*; *Boettiger et al., 2016*; *Shachar et al., 2015*). Recent advancements in DNA FISH (fluorescence in situ hybridization) have enabled high-throughput imaging of thousands of loci simultaneously (*Su et al., 2020*; *Takei et al., 2021*). These imaging studies have not only confirmed the structural features observed through sequencing techniques but have also provided valuable insights into the heterogeneity present at the single-cell level.

The abundance of available experimental data in the field of nuclear organization provides a fertile ground for structural modeling (*Qi et al., 2020*; *Qi and Zhang, 2019*; *Boninsegna et al., 2022*; *Fujishiro and Sasai, 2022*; *Shi and Thirumalai, 2021*; *Dekker et al., 2013*; *Jost et al., 2014*; *Giorgetti et al., 2014*; *Di Pierro et al., 2017*; *Buckle et al., 2018*; *Nuebler et al., 2018*; *Bianco et al., 2018*; *Shi et al., 2018*; *MacPherson et al., 2018*; *Shin et al., 2023*; *Amiad-Pavlov et al., 2021*; *Brahmachari et al., 2022*; *Jiang et al., 2022*; *Ganai et al., 2014*; *Liu et al., 2018*; *Laghmach et al., 2020*; *Chu and Wang, 2021*; *Lappala et al., 2021*; *Chu and Wang, 2022*; *Goychuk et al., 2023*; *Sun et al., 2021*; *Kadam et al., 2023*). To make sense of this wealth of information, various computational approaches have been introduced, with polymer simulation approaches being extensively utilized. These simulation techniques aid in reconstructing structural ensembles that closely replicate experimental data, offering valuable insights into the mechanisms underlying chromosome folding. In recent studies, these approaches have also been employed to investigate the interplay between the genome and the nuclear lamina (*Bajpai et al., 2021*; *Kamat et al., 2023*; *Laghmach et al., 2021*; *Stephens et al., 2018*), as well as nucleoli (*Qi and Zhang, 2021*), shedding light on their dynamic relationships.

Despite the progress made in computational modeling, the absence of well-documented software with easy-to-follow tutorials pose a challenge. Many research groups develop their own independent software, which complicates cross-validation and hinders the establishment of best practices for genome modeling (*Fujishiro and Sasai, 2022*; *Yildirim et al., 2023*; *Oliveira Junior et al., 2021*). Moreover, comprehensive models of the entire nucleus, especially at high resolution, remain scarce. Addressing these limitations and fostering collaboration in the scientific community can be achieved through the development of open-source tools. By promoting transparency and accessibility, such tools have the potential to greatly facilitate nuclear modeling and contribute to a more unified and collaborative research environment.

We present OpenNucleome, an open-source software designed for conducting molecular dynamics (MD) simulations of the human nucleus. This software streamlines the process of setting up whole nucleus simulations through just a few lines of Python scripting. OpenNucleome can unveil intricate, high-resolution structural and dynamic chromosome arrangements at a 100KB resolution. It empowers researchers to track the kinetics of condensate formation and fusion while also exploring

the influence of chemical modifications on condensate stability. Furthermore, it facilitates the examination of nuclear envelope deformation's impact on genome organization. The software's modular architecture enhances its adaptability and extensibility. Leveraging the power of OpenMM (*Eastman et al., 2017*), a GPU-accelerated MD engine, OpenNucleome ensures efficient simulations.

Our work demonstrates the fidelity of the simulated nuclear organizations by faithfully reproducing Hi-C, Lamin B DamID, TSA-Seq, and DNA-MERFISH data. The dynamic insights extracted from this model are pivotal in advancing our understanding of nuclear organization mechanisms. Our findings reveal that inherent heterogeneity in chromosome contacts naturally emerges within single cells. Interestingly, robust contacts between chromosomes and nuclear bodies can also be established due to a coupled self-assembly mechanism. Notably, the resilience of contacts involving nuclear bodies supports a nuclear zoning model for genome function. In the realm of nuclear investigations, we anticipate OpenNucleome to serve as an invaluable tool, seamlessly complementing experimental techniques.

## Results

### Non-equilibrium nucleus model at 100 KB resolution

We present an open-source implementation of a computational framework that facilitates the structural and dynamical characterization of the human nucleus. This framework builds upon a previous investigation but incorporates several significant modifications. Firstly, we enhance the model resolution by a factor of 10, enabling the precise determination of the spatial positioning of each chromatin segment measuring 100KB in length. Secondly, we present a kinetic scheme for speckles that accounts for the phosphorylation of protein molecules. This inclusion captures the influence of chemical reactions on the stability and dynamics of nuclear bodies. Thirdly, we incorporate explicit nuclear envelope dynamics to explore the impact of large-scale deformations on genome organization. Finally, our implementation into OpenMM offers the advantages of Python Scripting and GPU acceleration, facilitating easy extension and customization. These features will facilitate the broad applicability and adoption of the proposed model.

The nucleus model provides particle-based representations for chromosomes, nucleoli, speckles, and the nuclear envelope. As shown in *Figure 1A and B*, each of the 46 chromosomes is represented as a beads-on-a-string polymer, where each bead represents a 100-KB-long genomic segment. Based on Hi-C data, we further assign each bead as compartment *A*, *B*, or *C* to signify euchromatin, heterochromatin, or pericentromeric regions. The lamina was modeled as a spherical enclosure with 10 μm diameter, using discrete particles arranged to represent a mesh grid with covalent bonds linking together nearest neighbors (*Strom et al., 2021*). We modeled nucleoli and speckles as liquid droplets that emerge through the spontaneous phase separation of coarse-grained particles, representing protein and RNA molecule aggregates (*Chen and Belmont, 2019*; *Lafontaine et al., 2021*). These particles exhibited attractive interactions within the same type to promote condensation. More details about the various components of the system can be found in the Appendix 1, section 'Components of the whole nucleus model'.

The energy function of the nucleus model includes three components that account for the self-assembly of chromosomes, the assembly of nuclear bodies, and the coupling between chromosomes and nuclear landmarks. Therefore, the model approximates nuclear organization as a coupled self-assembly process. The chromosome energy function (see *Equation 7* in Appendix 1, section 'Hi-C inspired interactions for the diploid human genome') includes terms that account for the polymer connectivity and excluded volume effect, an ideal potential, compartment-specific interactions, and specific interchromosomal interactions. As shown in *Figure 1C*, the ideal potential is only applied for beads from the same chromosome to approximate the effect of loop extrusion by Cohesin molecules (*Sanborn et al., 2015*; *Fudenberg et al., 2016*) for chromosome compaction and territory formation (*Di Pierro et al., 2016*; *Zhang and Wolynes, 2017*). Compartment-specific interactions, on the other hand, promote microphase separation and compartmentalization of euchromatin and heterochromatin. Finally, interchromosomal interactions account for sequence-specific effects that compartment-dependent potentials cannot capture.

Interactions among coarse-grained particles that form nuclear bodies were designed to promote and stabilize the formation of liquid droplets, as has been revealed by many experiments (*Handwerger*

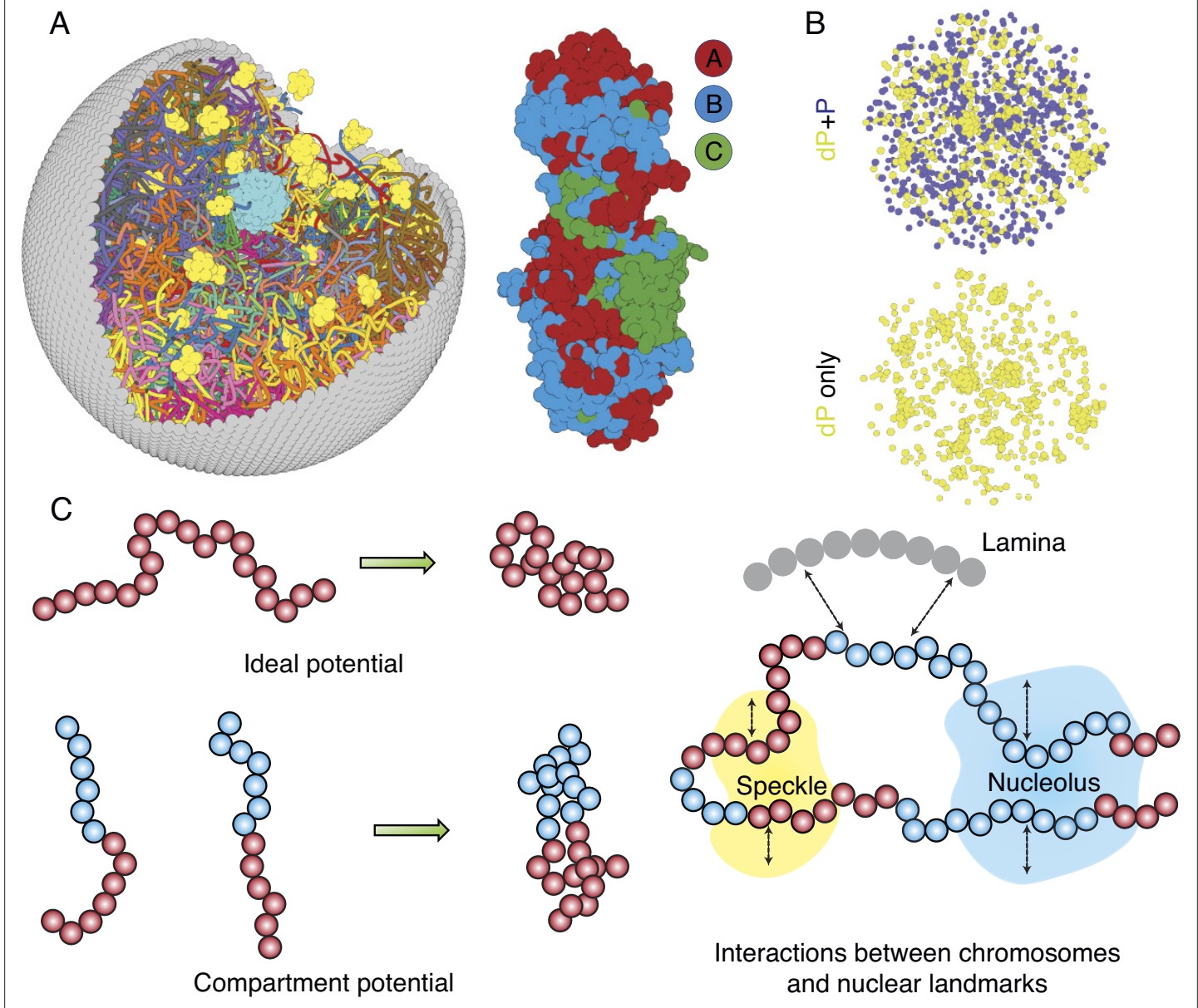

**Figure 1.** Computer model of the human nucleus for structural and dynamical characterizations. (**A**) 3D rendering of the nucleus model with particle-based representations for the 46 chromosomes shown as ribbons, the nuclear lamina (gray), nucleoli (cyan), and speckles (yellow). As shown on the right, chromosomes are modeled as beads-on-a-string polymers at a 100 KB resolution, with the beads further categorized into compartment A (red), compartment B (light blue), or centromeric regions (green). (**B**) Speckle particles undergo chemical modifications concurrent to their spatial dynamics, and the de-phosphorylated (dP) particles contribute to droplet formation. (**C**) Illustration of the ideal and compartment potential that promotes chromosome compaction and microphase separation. Specific interactions between chromosomes and nuclear landmarks are shown on the right.

_et al., 2005_; _Caragine et al., 2018_; _Caragine et al., 2019_). We adopted the Lennard–Jones potential for nucleolar particles to mimic the weak, multivalent interactions that arise from protein and RNA molecules that make up the nucleoli. As a first attempt to approximate their complex dynamics, we considered two types of particles that form speckles: phosphorylated (P) and de-phosphorylated (dP). The two types can interconvert via chemical reactions (_Brackley et al., 2017_; _Söding et al., 2020_; _Carrero et al., 2006_) and dP particles share attractive interactions modeled with the Lennard–Jones potential.

As shown in _Figure 1C_, to recognize specific interactions between chromosomes and nuclear landmarks, we introduced contact potentials between them. These potentials are inspired by the

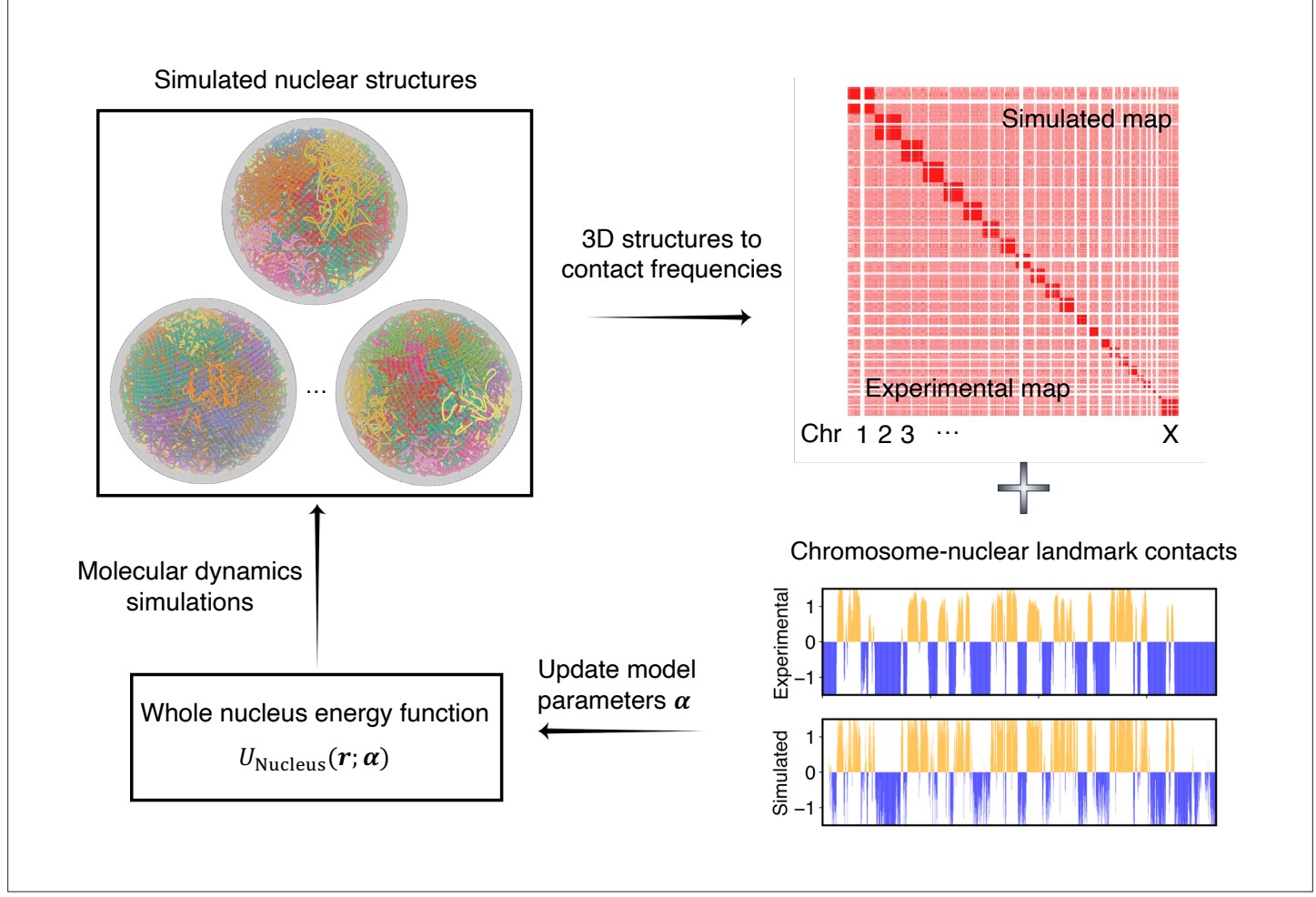

**Figure 2.** Overview of the iterative algorithm for parameterizing the nucleus model with experimental data. Starting from an initial set of parameters, we perform molecular dynamics (MD) simulations to produce an ensemble of nuclear structures. These structures can be transformed into contacts between chromosomes or between chromosomes and nuclear landmarks for direct comparison with experimental data. Differences between simulated and experiment contacts are used to update parameters for additional rounds of optimization if needed.

The online version of this article includes the following figure supplement(s) for figure 2:

**Figure supplement 1.** The number of speckle clusters formed along a typical simulation trajectory.

experimental techniques that probe the corresponding contacts. Appendix 1, sections 'Chromosome–nuclear landmark interactions' and 'Nuclear landmark–nuclear landmark interactions' contain more details about all the nuclear landmark-related energy functions.

## Optimization of model parameters with experimental data

The nucleus model was designed to be interpretable such that energy terms represent physical processes. Furthermore, the expressions of the interaction potentials were also designed such that their parameters can be determined from experimental data via the maximum entropy optimization algorithm (*Lin et al., 2021*; *Xie and Zhang, 2019*; *Schuette et al., 2023*). Below, we briefly outline the procedure used for parameter optimization and further details can be found in Appendix 1, section 'Optimization of the whole nucleus model parameters.

As illustrated in *Figure 2*, starting from a given set of parameters, we first perform MD simulations to produce a collection of 3D structures for the diploid genome and various nuclear bodies. These structures are then transformed into a contact map or contact probabilities between chromatin beads and nuclear landmarks by averaging over homologous chromosomes. Constraints corresponding to different energy terms could be obtained from the simulated results and compared with those

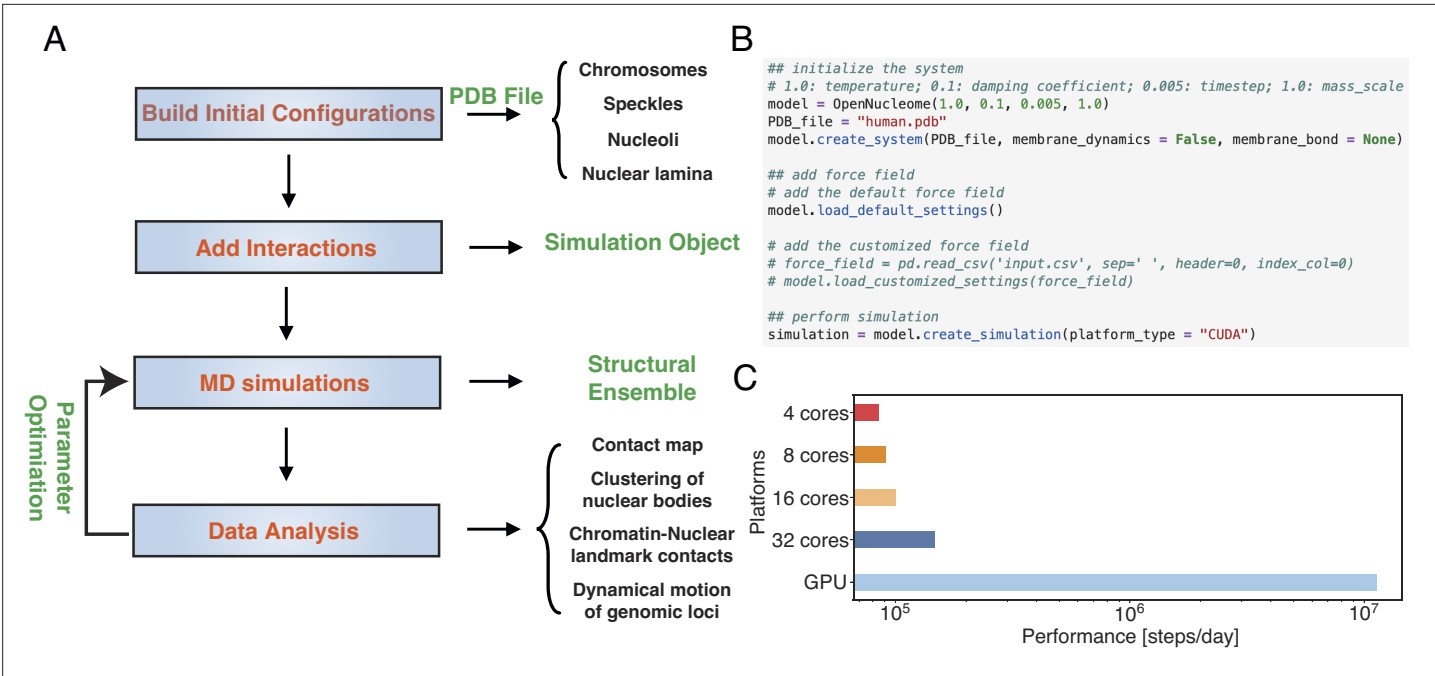

**Figure 3.** OpenNucleome facilitates GPU-accelerated simulations of the human nucleus. (**A**) Illustration of workflow for setting up, performing, and analyzing molecular dynamics (MD) simulations. (**B**) Python scripts setting up whole nucleus simulations. (**C**) Performance of MD simulations on different number of CPU cores and a single GPU.

estimated from Hi-C, SON TSA-Seq, and Lamin B DamID profiles. Finally, the model parameters were updated based on the difference between simulated and experimental constraints using the adaptive moment estimation (Adam) optimization algorithm (*Kingma and Ba, 2014*). The three steps can be repeated with updated parameters to improve the simulation-experiment agreement further.

No quantitative experimental data exists for interactions among nuclear body particles to serve as constraints. We varied the strength of the interaction potential to produce 2–3 nucleoli and ~30 speckle clusters during the simulations (*Figure 2—figure supplement 1*) while ensuring the fluidity of the resulting droplets.

## Molecular dynamics simulations with GPU acceleration

We implemented the nucleus model into the MD engine OpenMM (*Eastman et al., 2017*). OpenMM offers an excellent interface with Python scripting, significantly improving the readability and customizability of the model. The code was designed into functional modules, with different components, such as chromosomes and nuclear landmarks, written as separate classes. This design further facilitates the introduction of additional nuclear components, if desired, with minimal changes to existing code. We provide examples of simulation set up, trajectory analysis, parameter optimization, and introducing new features in the GitHub repository.

*Figure 3A* illustrates the workflow for setting up and executing whole nucleus simulations. A configuration file that provides the position of individual particles in the PDB file format is needed to initialize the simulations. This file also contains topological information regarding whether a particle represents chromosomes or nuclear landmarks and the identity of specific chromosomes. The input file can be generated with provided Python scripts by randomly distributing the positions of chromosomes, speckles, and nucleoli, though optimized configurations are also included in the GitHub repository. By default, the lamina particles will be uniformly placed on a sphere of 10 μm in diameter. Upon parsing the configuration file, interactions among various components can be set up with optimized parameters. This step will produce an object that can be used for MD simulations. As shown in *Figure 3B*, the workflow only requires a few lines of code. The package also includes analysis scripts to compute contact maps, monitor conformational dynamics, and track nuclear bodies.

A significant benefit of OpenMM is its native support of GPU acceleration. As shown in *Figure 3C*, the simulation speed with one Nvidia Volta V100 GPU is 150 times faster than that of the four Intel Xeon Platinum 8260 CPU cores. Notably, this performance enhancement cannot be achieved by simply increasing the CPU core numbers. For example, the simulation speed with 32 CPU cores is less than twice that of 4 CPU cores, potentially due to the system's heterogeneous distribution of particles.

## Simulations reproduce and predict diverse experimental data

We extensively validated the parameterized nucleus model to examine its biological relevance. MD simulations initialized from 50 different initial configurations were performed to build an ensemble of structures. As mentioned in the following section, a diverse set of initial configurations is essential for reproducing interchromosomal contacts probed in Hi-C. From the simulated structures, we computed various quantities for direct comparison with experimental measurements. Given that the majority of experimental data were analyzed for the haploid genome, we adopted a similar approach by averaging over paternal and maternal chromosomes to facilitate direct comparison. More details on data analysis can be found in Appendix 1, section 'Details of simulation data analysis'.

We compared the simulated contact probabilities among chromosomes with Hi-C data. As shown in *Figure 4A* and *Figure 4—figure supplement 1*, the simulated and experimental contact maps are highly correlated. The squares along the diagonal support the formation of chromosome territories that promote intrachromosomal contacts, and the apparent checkboard patterns follow the compartmentalization of various chromatin types. We further examined the decay of intrachromosomal contacts as a function of the sequence separation, which is known to deviate from that of an equilibrium globule (*Lieberman-Aiden et al., 2009*). As shown in *Figure 4B*, the simulated results overlap well with the Hi-C data (orange curve). In addition, the simulated average contact probabilities between various compartment types match values estimated from Hi-C data. Moreover, the simulated and experimental average contact probabilities between pairs of chromosomes agree well, and the Pearson correlation coefficient between the two datasets reaches 0.89.

We further examined the contacts between chromosomes and nuclear landmarks. As illustrated in *Figure 4C*, the simulated Lamin-B DamID signals for chromosome 7 match well with the experimental results, capturing the complex contact pattern that weaves chromatin toward and away from the nuclear envelope. Similarly, SON TSA-Seq data that quantify the contact between chromosomes and speckles are well captured by simulated structures. The anti-correlation between DamID and TSA-Seq is clearly visible. The observed agreement between simulation and experimental results is not limited to any particular chromosome. Good agreements are achieved for all chromosomes.

The simulations also provide 3D representations of the nucleus that can be compared with DNA-MERFISH data (*Su et al., 2020*). We found that the simulated radius of gyration of individual chromosomes matches well with experimental values (*Figure 4A*). The simulated and experimental average normalized chromosome radial positions also correlate strongly, as shown in *Figure 5B*. We note that while the sequencing results presented in *Figure 4* were used for model parameterization, the MERFISH data were not. Therefore, the simulation results here are de novo predictions, and their agreement with experimental data strongly supports the coupled assembly mechanism used for designing the energy function.

A significant advantage of MD simulation-based models is the dynamical information they naturally produce. We measured the dynamics of telomeres by tracking the mean-square displacements (MSDs), $\langle \mathbf{r}^2(\Delta t) \rangle$, as a function of time. In *Figure 5C*, we plot representative MSD trajectories over a 1-hr timescale. In line with previous research (*Di Pierro et al., 2018*; *Bronstein et al., 2009*; *Lee et al., 2021*), telomeres display anomalous subdiffusive motion. When fitted with the equation $\langle \mathbf{r}^2(\Delta t) \rangle = D_\alpha \Delta t^\alpha$, these trajectories yield a spectrum of $\alpha$ values, with a peak around 0.59. The exponent and the diffusion coefficient $D_\alpha = (27 \pm 11) \times 10^{-4} \mu m^2 \cdot s^{-\alpha}$ both match well with the experimental values (*Bronshtein et al., 2015*; *Jack et al., 2022*), upon setting the nucleoplasmic viscosity as $1 Pa \cdot s$ (see Appendix 1, section 'Mapping the reduced time unit to real time' for more details).

The good agreement in the dynamics of individual loci further inspired us to examine the diffusion of whole chromosomes. In particular, we plotted the normalized chromosome radial positions as a function of time in *Figure 6A*. Remarkably, we found that chromosomes appear arrested and no significant changes in their positions are observed over timescales comparable to the cell cycle (see

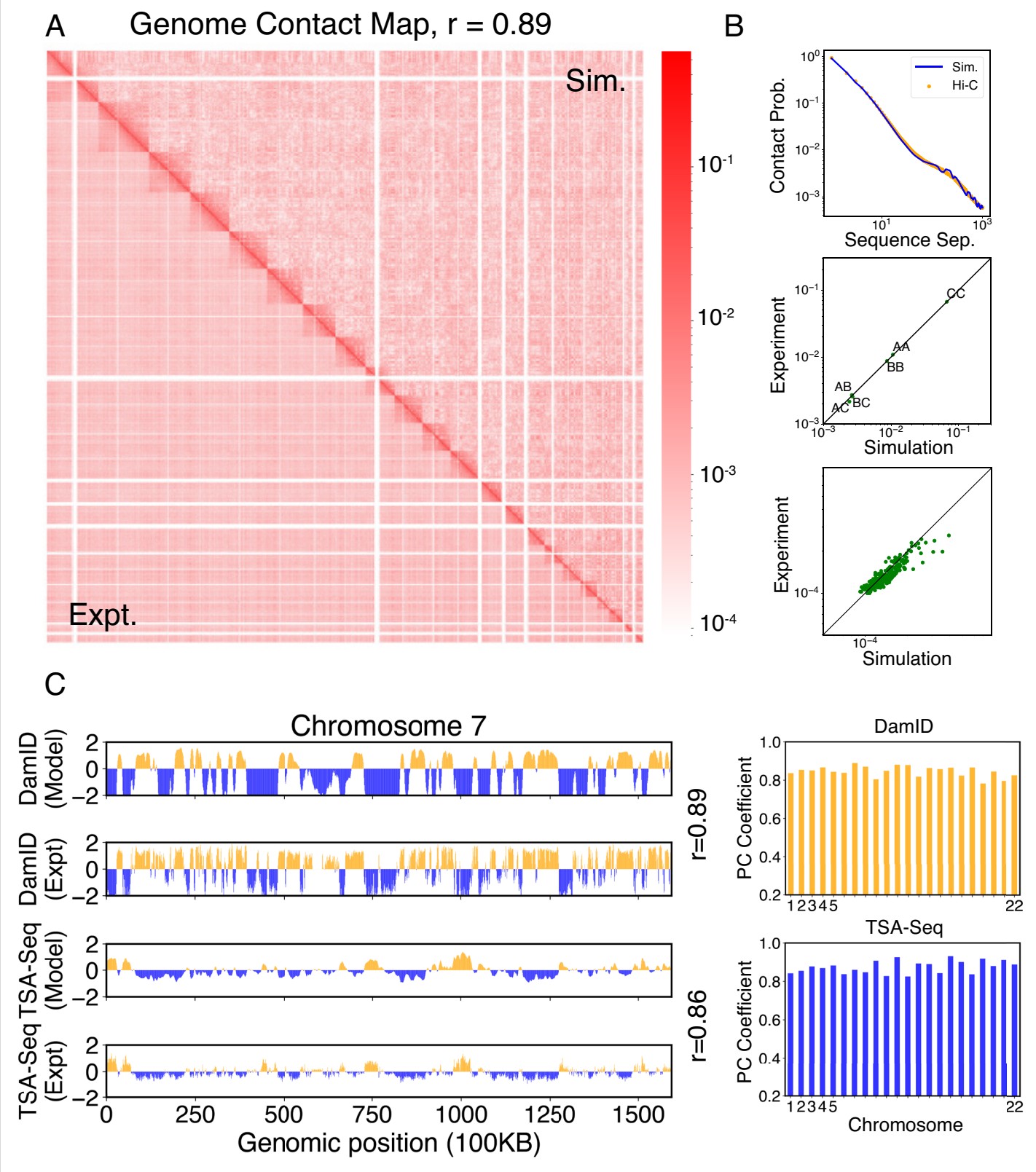

**Figure 4.** Simulated structures reproduce contact frequencies between chromosomes and between chromosomes and nuclear landmarks. (**A**) Comparison between simulated (top right) and experimental (bottom left) whole-genome contact probability maps with Pearson correlation coefficient r = 0.89. Zoom-ins of various regions are provided in *Figure 4—figure supplement 1*. (**B**) Comparison between simulated and experimental average contact frequencies, including average contacts between genomic loci from the same chromosomes at a given separation (top), average

*Figure 4 continued on next page*

*Figure 4 continued*

contacts between genomic loci classified into different compartment types (middle), and average contacts between various chromosome pairs (bottom). (**C**) Comparison between simulated and experimental Lamin-B DamID (top) and SON TSA-Seq signals (bottom), with Pearson correlation coefficients of haploid chromosomes shown on the right.

The online version of this article includes the following figure supplement(s) for figure 4:

**Figure supplement 1.** Zoom-in of various regions in the contact map presented in *Figure 4* further supports the agreement between simulation and experiment.

---

also *Figure 6—figure supplement 1*). Therefore, our simulations predict that large-scale movements of chromosomes are unlikely during the G1 phase.

## Heterogeneity and robustness of the simulated conformational ensemble

The lack of relaxation of chromosome radial positions suggests the importance of starting configurations used to initialize the simulations. Statistical averages of the resulting ensemble of nuclear structures depend crucially on these starting configurations. Using an optimization procedure, we selected them from 1000 configurations to maximize the agreement with experimental lamin-B DamID and interchromosomal contact probabilities. Appendix 1, section 'Initial configurations for simulations' provides more details on preparing the 1000 initial configurations.

We selected a total of 50 starting configurations to initiate independent simulations. Smaller sets of starting configurations are not sufficient to reproduce the interchromosomal contact probabilities, as shown in *Figure 6—figure supplement 2B*. Notably, different sets of 50 configurations selected from independent trials show significant overlap (*Figure 6—figure supplement 2D*), supporting the robustness of the selection protocol in detecting conserved features of genome organization.

While the ensemble as a whole is relatively robust, individual configurations with the ensemble exhibit significant differences. For example, the Lamin B DamID profiles produced from different trajectories are only weakly correlated (*Figure 6C*), with an average correlation coefficient of 0.53. These weak correlations result from significant differences in the normalized radial positions of chromosomes, as can be seen in representative configurations from two simulation trajectories (*Figure 6B*). The fluctuations of normalized radial positions cause changes in contacts between chromosomes as well, resulting in little correlation between interchromosomal contact matrices (*Figure 6D*).

We examined genome organizations reported by Su et al. and found a similar variation of interchromosomal contact probabilities across individual cells (*Figure 6—figure supplement 2A and D*). Notably, the simulated configurations capture the fluctuations of interchromosomal contacts observed in DNA-MERFISH data, further supporting the biological relevance of the reported in silico structures.

Despite the differences in interchromosomal contacts across trajectories, high conservation of connections between chromosomes and speckles can be observed in individual simulations. For example, the average correlation coefficient between in silico SON TSA-Seq profiles produced from different trajectories is 0.72, much higher than the corresponding value for Lamin B DamID profiles. Conservation of contacts between chromosomes and nuclear bodies (zones) across individual cells has indeed been reported in a previous study that simultaneously images chromatin and various subnuclear structures (*Takei et al., 2021*).

## Nuclear deformation preserves chromosome–nuclear body contacts

Numerous studies have highlighted the remarkable influence of nuclear shape on the positioning of chromosomes and the regulation of gene expression (*Brahmachari et al., 2022*; *Contessoto et al., 2023*). The nucleus, once regarded as a mere compartment for DNA storage, is increasingly recognized as a dynamic and intricately structured organelle. To better understand the interplay between nuclear shape and genome organization as a fundamental mechanism that shapes the transcriptional landscape, we performed additional simulations in which the nuclear lamina was altered from a sphere into more ellipsoidal shapes by applying a force along the *z*-axis (*Figure 7A*). More details about these simulations can be found in Appendix 1, section 'Nuclear envelope deformation simulations'.

As illustrated in *Figure 7B*, the presence of external forces resulted in significant alterations in nuclear shape. We conducted two independent simulations with different force strengths, leading

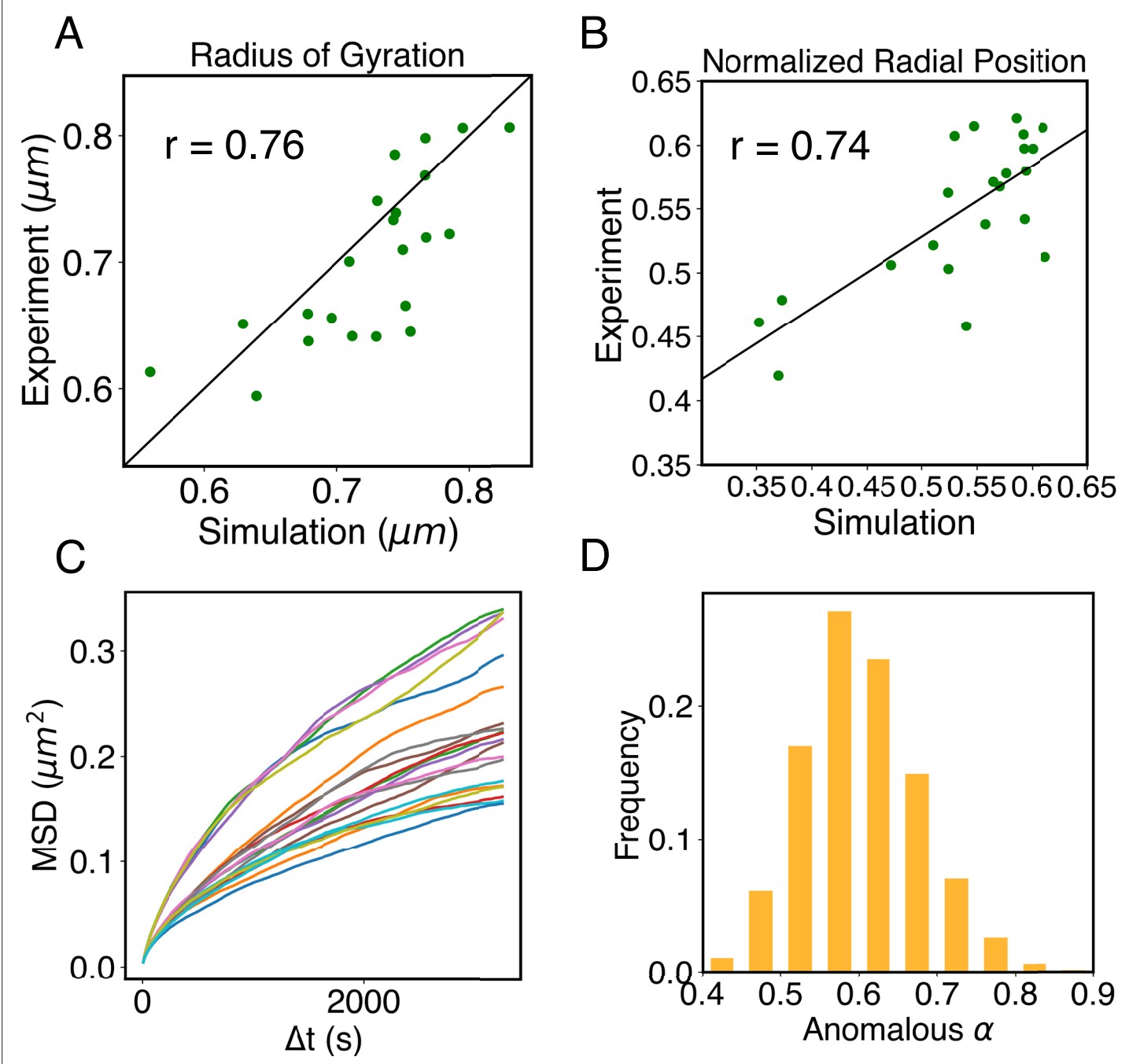

**Figure 5.** Structural and dynamical predictions of the nucleus model match results from microscopy imaging. (**A**) Comparison between the simulated and experimental radius of gyration, $R_g$, for haploid chromosomes. The Pearson correlation coefficient between the two, $r$, is shown in the legend. (**B**) Comparison between the simulated and experimental normalized radial positions for haploid chromosomes, with their Pearson correlation coefficient shown in the legend. Detailed definition of the normalized radial positions is provided in Appendix 1, section 'Computing simulated normalized chromosome radial positions'. (**C**) Mean-squared displacements (MSDs) as a function of time are shown for selected telomeres. (**D**) The probability distribution of the anomalous exponent, $\alpha$, obtained from fitting the MSDs curves for all telomeres with the expression, $\langle \mathbf{r}^2 (\Delta t) \rangle = D_\alpha \Delta t^\alpha$.

to varying degrees of deformation in the nuclear lamina. This deformation, in turn, caused a reorganization of chromosomes, affecting their normalized radial positions and pairwise contacts (see *Figure 7—figure supplement 1* and *Figure 7C*). We observed that more deformed nuclei exhibited lower correlation coefficients for interchromosomal contacts compared to results obtained from simulations in a spherical nucleus. Similarly, the DamID profiles exhibited significant variations upon nucleus

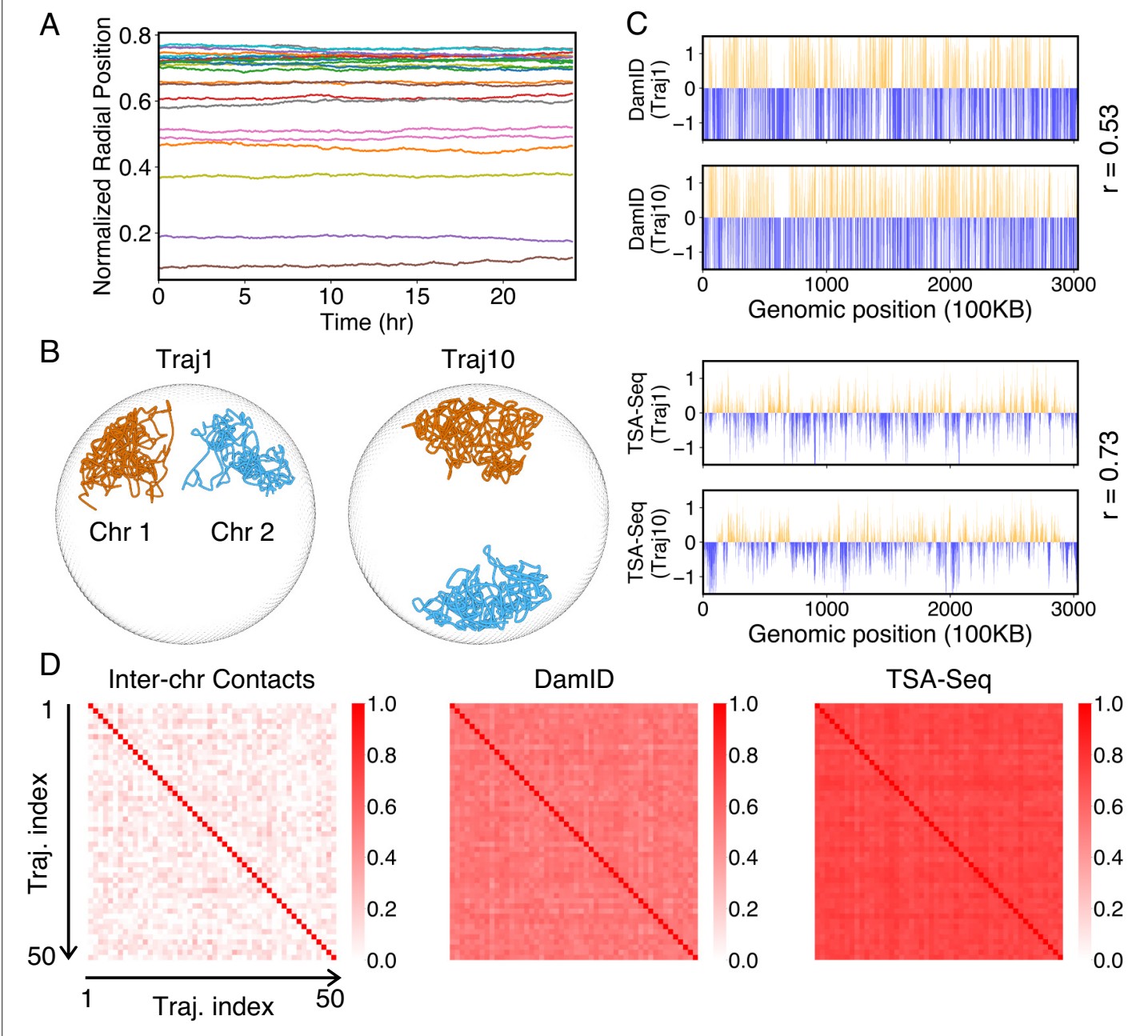

**Figure 6.** Heterogeneity and conserved features of nuclear organizations. (**A**) Normalized chromosome radial positions as a function of simulation time. (**B**) Contacts between chromosomes 1 and 2 from two independent simulation trajectories show significant variations. (**C**) Genome-wide in silico Lamin B DamID (top) and SON TSA-Seq (bottom) profiles computed from two independent trajectories. Pearson correlation coefficients, *r*, are provided on each plot. (**D**) Pairwise Person correlation coefficients between interchromosomal contact matrices (left), genome-wide Lamin B DamID profiles (middle), and genome-wide SON TSA-Seq profiles (right) determined from independent trajectories. The averages excluding the diagonals of the three datasets are 0.06, 0.53, and 0.72.

The online version of this article includes the following figure supplement(s) for figure 6:

**Figure supplement 1.** Arrested kinetics of chromosome positions over the timescale of cell cycles.

**Figure supplement 2.** Configurations used to initialize simulations capture the heterogeneity in interchromosomal contacts seen in DNA-MERFISH data.

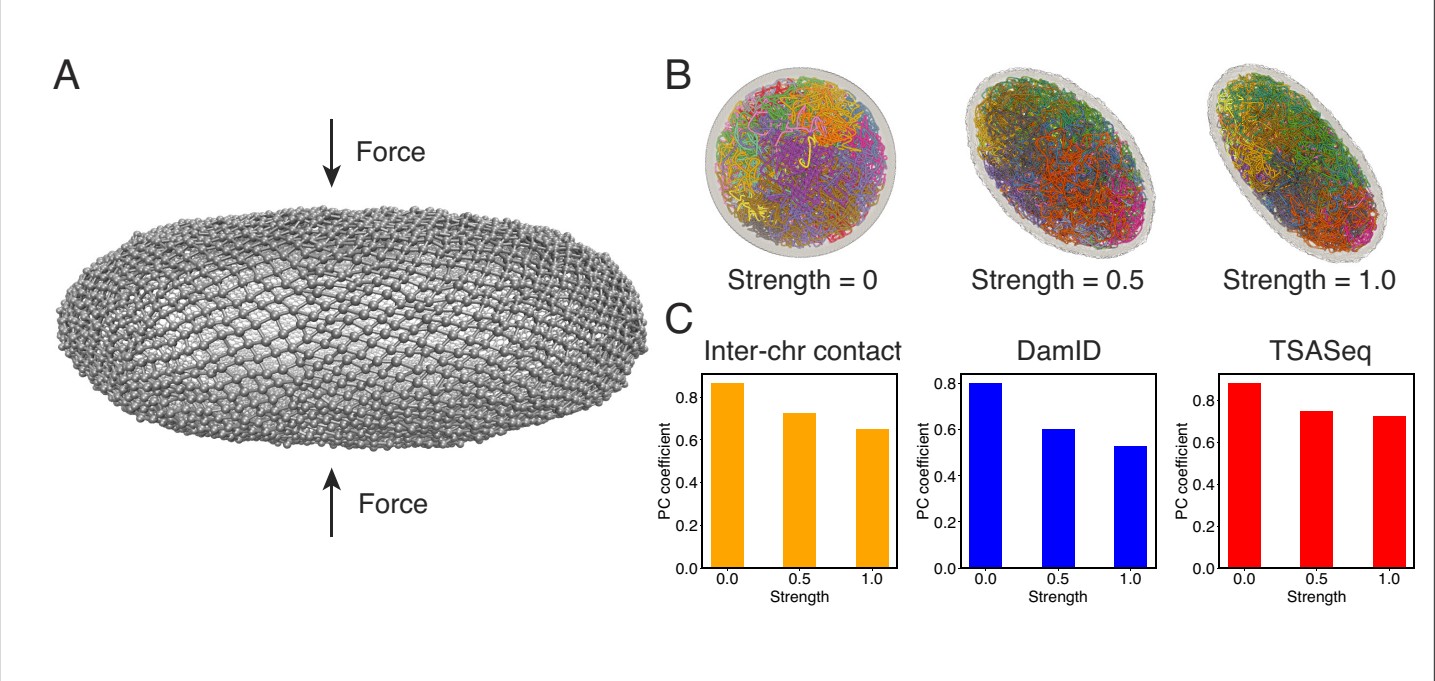

**Figure 7.** Nuclear deformations influence genome organization while preserving chromatin-speckle contacts. (**A**) Illustration of force-induced nuclear envelope deformation. The nuclear lamina is modeled as a particle mesh where neighboring lamina particles are covalently bonded together. (**B**) Example nucleus conformations at different strengths of applied force. (**C**) Pearson correlation coefficients between results from simulations of deformed nuclei and those from a spherical nucleus for interchromosomal contacts (left), DamID profiles (middle), and TSA-Seq (right). The values at zero force were computed from two independent simulations starting from the same initial configurations.

The online version of this article includes the following figure supplement(s) for figure 7:

**Figure supplement 1.** Impact of nuclear deformation on normalized chromosome radial positions.

deformation, whereas TSA-Seq signals were much less affected and remained highly correlated with the results from the spherical nucleus simulations.

Therefore, it appears that speckles, and potentially other nuclear condensates, can dynamically reorganize in response to changes in chromosome conformations to maintain contacts with genomic loci. This robustness in nuclear body contacts may be essential for ensuring the robust functioning of the genome in a population of cells with significant variability in nuclear shape.

## Discussion

We introduced a computational model, OpenNucleome, to facilitate simulations for the human nucleus at high structural and temporal resolution. We conducted extensive cross-validation with experimental data to support the biological relevance of simulated 3D structures. Implementing the model into the MD package, OpenMM enables GPU acceleration for long-timescale simulations. Tutorials in the format of Python Scripts with extensive documentation are provided to facilitate the adoption of the model by the community.

Our software enhances the capabilities of existing genome simulation tools *Fujishiro and Sasai, 2022*; *Yildirim et al., 2023*; *Oliveira Junior et al., 2021*. Specifically, OpenNucleome aligns with the design principles of Open-MiChroM (*Oliveira Junior et al., 2021*), prioritizing open-source accessibility while expanding simulation capabilities to the entire nucleus. Similar to software from the Alber lab (*Yildirim et al., 2023*), OpenNucleome offers high-resolution genome organization that faithfully reproduces a diverse range of experimental data. Furthermore, beyond static structures, OpenNucleome facilitates dynamic simulations with explicit representations of various nuclear condensates, akin to the model developed by *Fujishiro and Sasai, 2022*.

A significant advantage of OpenNucleome lies in its predictive power for dynamical information. For example, the model succeeded in reproducing the subdiffusive behavior of telomeres. We further

showed that the dynamics of individual chromosomes are slow and their radial positions do not relax over the time course of a cell cycle. This is consistent with previous theoretical estimations on chromosome dynamics (*Rosa and Everaers, 2008*) and recent observations of solid behavior of chromatin in vivo (*Strickfaden et al., 2020*). Live cell experiments that directly track the positions of multiple chromosomes could further validate/falsify this prediction. We anticipate the model will greatly facilitate the investigation of the dynamics of genomic loci and nuclear bodies and the interpreting of live cell imaging results.

Slow chromosome dynamics and a lack of conformational relaxation naturally result in the heterogeneity of chromosome radial positions across individual cells. This heterogeneity raises doubts about the notion that chromosome radial positions provide robust and reliable mechanisms for gene regulation (*Hübner et al., 2013*; *Maeshima et al., 2010*; *Fraser and Bickmore, 2007*; *Takizawa et al., 2008*). Instead, our results support the nuclear zoning model for gene regulation (*Takei et al., 2021*), where specific loci function as 'fixed points' anchored to certain nuclear bodies in all cells. This anchoring mechanism robustly creates the desired molecular environment surrounding these genomic segments. Unlike chromosome radial positions, contacts between genomic loci and speckles can be robustly established in individual cells, as shown in our simulations. It was achieved through a nucleation process that attracts speckle particles toward specific loci due to specific interactions. Nucleation occurs much more rapidly than chromosome rearrangement due to the smaller size of speckle particles. The coupled self-assembly mechanism for chromosomes and nuclear bodies can similarly facilitate the formation of other nuclear zones for different kinds of fixed points.

Despite the heterogeneity in chromosome positions and interchromosomal contacts, the ensemble of nuclear structures as a whole is not random and exhibits conserved features. For example, on average, certain chromosomes remain closer to the nuclear envelope than others (see *Figure 5B*). Similarly, the average contact frequency between certain chromosome pairs is higher than others, though this trend can be frequently violated in individual cells. How such conserved features arise as cells exit from the mitotic phase remains unclear and would be interesting for further explorations.

## Methods
### Molecular dynamics simulation details

We used the software package OpenMM *Eastman et al., 2017* to perform MD simulations in reduced units at constant temperature ($T = 1.0$). Unless otherwise specified, we froze the lamina particles and only propagated the dynamics of chromatin, nucleoli, and speckles.

Two integration schemes were used with a time step of $dt = 0.005$ to efficiently generate structural ensembles and produce realistic dynamical information, respectively. For simulations used in parameter optimization and building structural ensembles, we employed the Langevin integrator with a damping coefficient of $\gamma^{-1} = 10.0$. In the case of MSD calculations shown in *Figure 6*, we utilized Brownian dynamics with a damping coefficient of $\gamma^{-1} = 0.01$. The higher damping coefficient provides a better approximation to the viscous nucleus environment, while the smaller value in the Langevin integrator facilitates conformational sampling with faster diffusion rates.

We employed the semi-grand Monte Carlo technique (*Sadigh et al., 2012*) to simulate chemical transitions between two types of speckle particles. At every 4000 simulation steps, we attempt a total of $N_{Sp}$ chemical reactions that converts one type of speckle particles to the other type with a probability of 0.2. $N_{Sp}$ corresponds to the total number of speckle particles, and the switching probability was chosen to be comparable to the experimental phosphorylation rate. More details on the speckle dynamics are provided in Appendix 1, section 'Speckles as phase-separated droplets undergoing chemical modifications'.

When deforming the nuclear envelope, we unfroze the lamina particles and evolved them dynamically as the rest of the nucleus. Bonded interactions among lamina particles held the nuclear envelope together as a particle mesh. A harmonic force along the $z$-axis was introduced to compress the particle mesh. More details are provided in Appendix 1, section 'Nuclear envelope deformation simulations'.

For simulations used to optimize parameters, a total of 50 independent 3-million-step-long trajectories were performed. Configurations were recorded at every 2000 simulation steps for analysis. The first 500,000 steps of each trajectory were discarded as equilibration. For production simulations, we performed 50 independent 10-million-step long trajectories starting from different initial

configurations. Nuclear structures were again recorded at every 2000 steps to determine statistical averages presented in the article. An additional eight simulations of 30million steps in length were performed to compute telomere MSDs.

We mapped the reduced units to real units with the conversion of length scale $\sigma$ = 385nm and the timescale in Brownian dynamics simulations $\tau = 0.65s$. These conversions were determined as detailed in Appendix 1, section 'Unit conversion'.

## Experimental data processing and analysis

We obtained the in situ Hi-C data, SON TSA-seq data, and Lamin-B DamID data of HFF cell lines from the 4DN data portal. The intra and interchromosomal interactions were calculated at 100KB resolution with VC_SQRT normalization applied to the interaction matrices. Hi-C data extraction and normalization were performed using Juicer tools (*Durand et al., 2016*). We followed the same processing and normalization method described in *Zhang et al., 2021* to analyze TSA-seq data. Two biological replicates of Lamin-B DamID data were merged and the normalized counts over Dam-only control were used for analysis. The SON TSA-Seq and Lamin-B DamID data were processed at the 25KB resolution and the average values at the 100KB resolution were used in *Figure 4* for model validation.

## Acknowledgements

This work was supported by the National Institutes of Health (grant R35GM133580).

## Additional information

### Competing interests

Bin Zhang: Reviewing editor, *eLife*. The other authors declare that no competing interests exist.

### Funding

| Funder | Grant reference number | Author |
| --- | --- | --- |
| National Institute of General Medical Sciences | R35GM133580 | Bin Zhang |

The funders had no role in study design, data collection and interpretation, or the decision to submit the work for publication.

### Author contributions

Zhuohan Lao, Software, Formal analysis, Investigation, Visualization, Methodology, Writing - original draft, Writing - review and editing; Kartik D Kamat, Formal analysis, Methodology; Zhongling Jiang, Software; Bin Zhang, Conceptualization, Formal analysis, Supervision, Funding acquisition, Investigation, Visualization, Methodology, Writing - original draft, Project administration, Writing - review and editing

### Author ORCIDs

Zhuohan Lao http://orcid.org/0000-0001-5404-2183
Bin Zhang https://orcid.org/0000-0002-3685-7503

Reviewer #1 (Public Review): https://doi.org/10.7554/eLife.93223.3.sa1
Reviewer #2 (Public Review): https://doi.org/10.7554/eLife.93223.3.sa2
Reviewer #3 (Public Review): https://doi.org/10.7554/eLife.93223.3.sa3
Author response https://doi.org/10.7554/eLife.93223.3.sa4

## Additional files

### Supplementary files
• MDAR checklist

## Data availability

Hi-C data (https://data.4dnucleome.org, accession number: 4DNFIB59T7NN). SON TSA-seq data (https://data.4dnucleome.org, accession number: pulldown data 4DNEX6U8TS3Y, control data 4DNEXI7XUWFK). LaminB DamID data (https://data.4dnucleome.org, accession number 4DNESX-Z4FW4T). The software is available at https://github.com/ZhangGroup-MITChemistry/OpenNucleome (copy archived at *ZhangGroup-MITChemistry, 2024*).

The following previously published datasets were used:

| Author(s) | Year | Dataset title | Dataset URL | Database and Identifier |
|---|---|---|---|---|
| van Steensel B, NKI | 2017 | LaminB1 DamID of HFFc6 Tier 1 cells – cells were transduced with virus expressing Dam-LaminB1, gDNA was harvested after 4 days and processed for DamID-seq | https://data.4dnucleome.org/experiment-set-replicates/4DNESXZ4FW4T/ | 4DN Data Portal, 4DNESXZ4FW4T |
| Zhang L, Zhang Y, Chen Y, Gholamalamdari O, Wang Y, Ma J, Belmont AS | 2020 | Set of Input for SON Ab2 TSA-seq version 2 Reaction Condition 2 (PBS 50% Sucrose) Enhancement Condition E (1:300 tyramide-biotin, 30 minute reaction) on HFFc6 cells | https://data.4dnucleome.org/experiment-set-replicates/4DNESB5I8TGR | 4DN Data Portal, 4DNEXI7XUWFK |
| Gholamalamdari O, van Schaik T, Wang Y, Kumar P, Zhang L, Zhang Y, Hernandez Gonzalez GA, Vouzas AE, Zhao PA, Gilbert DA, Ma J, van Steensel B, Belmont AS | 2024 | TSA-seq against SON protein on HFFc6 (Tier 1) | https://data.4dnucleome.org/experiments-tsaseq/4DNEX6U8TS3Y | 4DN Data Portal, 4DNEX6U8TS3Y |
| Krietenstein N, Abraham S, Venev SV, Abdennur N, Gibcus J, Hsieh TS, Parsi KM, Yang L, Maehr R, Mirny LA, Dekker J, Rando OJ | 2020 | Ultrastructural Details of Mammalian Chromosome Architecture | https://data.4dnucleome.org/publications/a716e6b4-9cfa-4f8d-a2c7-cabf21d42b95 | 4DN Data Portal, 4DNFIB59T7NN |
| van Steensel B | 2017 | DamID-seq with DAM-LMNB1 on HFFc6 (Tier 1) | https://data.4dnucleome.org/experiments-damid/4DNEXFUGLVQA/ | 4DN Data Portal, 4DNEXFUGLVQA |

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

# Appendix 1

## Components of the whole nucleus model

As outlined in the main text, the whole nucleus model consists of chromosomes, nucleoli, speckles, and the nuclear lamina. Below, we provide details on the particle-based representations of the various components, totaling 70542 coarse-grained beads. Abbreviations are frequently used for clarity in notation, with N for nucleus, La for lamina, No for nucleoli, and Sp for speckles.

### Chromosomes as beads on the string polymers

We explicitly modeled the 46 human chromosomes as beads-on-a-string polymers. Each coarse-grained bead represents a 100 KB genomic segment, totaling 60642 beads for the genome. We assigned each bead as either compartment type *A*, *B*, *C*, or *N*. The compartment assignments for types *A* and *B* were extracted from the Hi-C contact matrix for HFF cells (*Krietenstein et al., 2020*) using the cooltools software (*Venev et al., 2020*), and compartment *C* were identified as centromeric regions based on the DNA sequence. Compartment *N* denotes genomic regions that cannot be assigned as *A*, *B*, or *C* due to a lack of Hi-C data.

### The nuclear lamina as a particle-based mesh

The nuclear envelope provides an enclosure to confine DNA and a repressive environment to organize chromatin with specific interactions (*Hetzer, 2010*). To account for the role of the nuclear lamina while keeping our model simple, we approximate it with discrete particles uniformly placed on a sphere.

Following our previous work (*Kamat et al., 2023*), we used the Fibonacci grid to initialize the lamina particles, which form a uniform and almost equidistant network of lamina particles on the surface of the nucleus (*Swinbank and James Purser, 2006*; *Li et al., 2007*). The Cartesian coordinates associated with the $i$th lamina particles are defined as

$$
\begin{aligned}
x_i &= 2R_{\mathrm{N}} \times \left( 1 - \frac{i}{N_{\mathrm{La}} - 1} \right) \\
y_i &= \sqrt{R_{\mathrm{N}}^2 - x^2} \times cos\left[ i\Phi \right] \\
z_i &= \sqrt{R_{\mathrm{N}}^2 - x^2} \times sin\left[ i\Phi \right]
\end{aligned}
\tag{1}
$$

where $N_{\mathrm{La}} = 8000$ represents the number of lamina particles, $i \in \{0, 1, \ldots, N_{\mathrm{La}} - 2, N_{\mathrm{La}} - 1\}$, and $\Phi = \pi \times \left( 3 - \sqrt{5} \right)$ is the golden angle. We set $R_{\mathrm{N}} = 5\mu\mathrm{m}$ as the radius of the human foreskin fibroblasts (HFF) cell nucleus.

### Nucleoli as phase-separated droplets

Nucleoli have been shown to behave as liquid droplets that form through phase separation (*Lafontaine et al., 2021*; *Pederson, 2011*; *Shin and Brangwynne, 2017*). We modeled the droplets with coarse-grained beads. While the composition of nucleoli is rather complex, we only used one type of particle for simplicity. In our simulations, we fixed the number of nucleolus particles, $N_{\mathrm{No}}$, based on the experimental concentration of nuclear protein NPM1, $c = 1\mu\mathrm{M}$ (*Qi and Zhang, 2021*; *Kamat et al., 2023*; *Zhu et al., 2019*). For example,

$$
N_{\mathrm{No}} = \frac{4\pi}{3} \cdot N_{\mathrm{A}} \cdot \left( R_{\mathrm{No}} \right)^3 \cdot c \approx 300
\tag{2}
$$

where $N_{\mathrm{A}}$ is the Avogadro constant and $R_{\mathrm{No}} = 0.5\mu\mathrm{m}$ is the average nucleolous size (*Caragine et al., 2018*; *Caragine et al., 2019*).

### Speckles as phase-separated droplets undergoing chemical modifications

Similar to nucleoli, speckles have also been shown to behave as liquid droplets (*Chen and Belmont, 2019*). However, one crucial unique feature of speckles is the constant chemical modifications of protein molecules comprising them, such as splicing factors (*Spector and Lamond, 2011*). The phosphorylation of these molecules has been argued to be essential for the dynamics and the number of speckles. Therefore, we implemented a kinetic scheme introduced by de Vries and coworkers to account for the chemical reactions. In this scheme, we consider two types of speckle

molecules: phosphorylated (Sp-P) and de-phosphorylated (Sp-dP). Only Sp-dP particles share attractive interactions.

The two protein types can inter-convert via chemical reactions with a transition probability matrix **T** defined as

$$
\mathbf{T} = \begin{array}{c} \\ S_p-dP \\ S_p-P \end{array} \begin{array}{cc} Sp-dP & Sp-P \\ \left\| \begin{array}{cc} p_{11} & p_{12} \\ p_{11} & p_{12} \end{array} \right\| \end{array} = \begin{array}{c} \\ S_p-dP \\ S_p-P \end{array} \begin{array}{cc} Sp-dP & Sp-P \\ \left\| \begin{array}{cc} 0.8 & 0.2 \\ 0.2 & 0.8 \end{array} \right\| \end{array} \tag{3}
$$

For simplicity, we assume the forward transition rate from Sp-P to Sp-dP particles is identical to the reverse rate. Because of the symmetry in transition rates, the average number of dP particles $\langle N_{\text{Sp-dP}} \rangle = 0.5 N_{\text{Sp}}$, where $N_{\text{Sp}}$ is the total number of speckle particles.

We chose the transition probability as 0.2 to be consistent with the phosphorylation rate. In particular, we estimate the rate as

$$
k_{12} = p_{12} \times \tau^{-1} = 0.2 \times \frac{1}{4000 \times 0.005 \times 0.65 s} = 0.0154 \, s^{-1} \tag{4}
$$

where $\tau$ is the time interval between consecutive attempts of chemical reactions. As detailed in section 'Molecular dynamics simulation details', the reactions were attempted every 4000 simulation steps, with a time step of 0.005. The time unit in our simulations is 0.65 s (see section 'Mapping the reduced time unit to real time'). The estimated value for $k_{12}$ is in the same order as the experimental phosphorylation rate (**Velazquez-Dones et al., 2005**).

We estimated the total number of speckle particles as follows. Assuming that there is a total of 30 speckles (**Galganski et al., 2017**), we have $N_{\text{Sp-dP}} = 30 \times N_s$, where $N_s$ is the number of Sp-dP particles in each cluster. This estimation assumes that only Sp-dP particles share attractive interactions and contribute to cluster formation. From the experimentally estimated relative mass densities of the protein concentrations in the speckle and nucleolus droplet as $\frac{170}{203}$ (**Handwerger et al., 2005**), we have

$$
\frac{N_s \times m/0.3^3}{100 \times m/0.5^3} = \frac{170}{203}. \tag{5}
$$

We assumed that speckle and nucleolus particles have identical mass and each nucleolus has 100 particles. The radius for speckle and nucleolus was approximated as 0.3 and $0.5\mu$m, yielding $N_s \approx 20$ and $N_{\text{Sp-dP}} \approx 600$. Because of the kinetic scheme defined in **Equation 3**, only parts of Sp-dP particles will participate in droplet formation during the simulations. Therefore, we increase the particle number and set $\langle N_{\text{Sp-dP}} \rangle = 800$, which yields $N_{\text{Sp}} = 1600$.

## Energy function of the whole nucleus model

As detailed below, the energy function of the whole nucleus, $U_{\text{Nucleus}}$, consists of interactions among chromosomes, among nuclear landmarks, and cross interactions between the two. Therefore,

$$
U_{\text{Nucleus}} = U_{\text{Genome}} + U_{\text{NL}} + U_{\text{GN}} \tag{6}
$$

## Hi-C inspired interactions for the diploid human genome

The energy function of the genome model is defined as

$$
U_{\text{Genome}} = U_{\text{homo}}(\mathbf{r}) + U_{\text{ideal}}(\mathbf{r}) + U_{\text{compt}}(\mathbf{r}) + U_{\text{inter}}(\mathbf{r}). \tag{7}
$$

$U_{\text{homo}}(\mathbf{r})$ determines a generic polymeric topology of chromosomes with excluded volume effect:

$$
U_{\text{homo}}(\mathbf{r}) = \sum_i \left[ u_{\text{bond}}(r_{i,i+1}) + u_{\text{angle}}\left( \vec{r}_{i,i+1}, \vec{r}_{i+1,i+2} \right) \right] + U_{\text{sc}}(\mathbf{r}) \tag{8}
$$

where the subscripts $i, i+1$, and $i+2$ represent the index of $i^{th}, (i+1)^{th}$, and $(i+2)^{th}$ beads, respectively, and $u_{\text{bond}}(r_{i,i+1})$ and $u_{\text{angle}}(r_{i,i+1}, r_{i+1,i+2})$ denote the bonding and angular potential applied for neighboring beads to ensure the connectivity of the chromatin chain and follow:

$$u_{\text{bond}}\left(r_{i,i+1}\right) = K_2\left(r - r_0\right)^2 + K_3\left(r - r_0\right)^3 + K_4\left(r - r_0\right)^4, K_2 = K_3 = K_4 = 20\epsilon$$

$$u_{\text{angle}}\left(\overrightarrow{r}_{i,i+1}, \overrightarrow{r}_{i+1,i+2}\right) = K_a\left[1 - \cos\left(\theta - \pi\right)\right], K_a = 2\epsilon, \cos\theta = \frac{\overrightarrow{r}_{i,i+1} \cdot \overrightarrow{r}_{i+1,i+2}}{|\overrightarrow{r}_{i,i+1}| \cdot |\overrightarrow{r}_{i+1,i+2}|} \tag{9}$$

where, as discussed in **Equation 32**, $r_0 = 0.5\sigma$ represents the size of the chromatin bead. The soft-core potential provides excluded volume effects for pairs of beads from the same or different chromosomes and follows:

$$U_{\text{sc}}\left(\mathbf{r}\right) = \sum_{j>i} u_{\text{sc}}\left(r_{ij}\right) \tag{10}$$

where $u_{\text{sc}}\left(r_{ij}\right)$ denotes a soft-core potential added to each pair formed by beads index $i$ and $j$ to account for the excluded volume effect while allowing the finite probability of cross-over of polymer chains.

$$u_{\text{sc}}\left(r_{ij}\right) = \begin{cases} 0.5E_{\text{cut}}\left(1 + \tanh\left[\frac{2U_{\text{LJ}}\left(r_{ij}\right)}{E_{\text{cut}}} - 1\right]\right), & r_{ij} \leq r_{\text{cut}} \\ U_{\text{LJ}}\left(r_{ij}\right), & r_{\text{cut}} < r_{ij} \leq 0.5 \times 2^{1/6}\sigma \\ 0, & r_{ij} > 0.5 \times 2^{1/6}\sigma \end{cases} \tag{11}$$

which corresponds to the Lennard–Jones potential capped off at a finite volume within a repulsive core to allow for chain crossing at a finite energy cost. $E_{\text{cut}} = 4\epsilon$ and $r_{\text{cut}}$ is chosen as the distance at which $U_{\text{LJ}}\left(r\right) = 0.5E_{\text{cut}}$.

$U_{\text{ideal}}\left(\mathbf{r}\right)$ is the intra-chromosomal potential applied to genomic loci within the same chromosome, while $U_{\text{compt}}\left(\mathbf{r}\right)$ is the compartment-specific interaction potential. The ideal potential, which can be rigorously derived following the maximum entropy principle (**Roux and Weare, 2013**; **Zhang and Wolynes, 2015**), adopts the following form:

$$U_{\text{ideal}}\left(\mathbf{r}\right) = \sum_{I}\sum_{i,j\in I}\alpha_{\text{ideal}}(|i - j|)f\left(r_{ij}\right) \tag{12}$$

where $I$ indexes over each chromosome and $i$ and $j$ index over pair of beads on that chromosome. $\alpha_{\text{ideal}}(|i - j|)$ depends only on the sequence separation between two beads $i$ and $j$. $f(r_{ij})$ measures the probability of contact formation for two loci separated by a distance of $r_{ij}$, and its ensemble average corresponds to the contact probability measured in Hi-C experiments. $f(r_{ij})$ adopts the form

$$\begin{aligned} f(r_{ij}) &= \frac{1}{2}(1 + \tanh[(r_c - r_{i,j})^{-5} + 5(r_c - r_{i,j})]) \\ &+ \frac{1}{2}(1 + \tanh[(r_c - r_{i,j})^{-5} + 5(r_c - r_{i,j})]) \times \left(\frac{r_c}{r_{i,j}}\right)^4 \end{aligned} \tag{13}$$

The numerical value of $r_c$ was determined from the Hi-C contact map, as detailed in the next section. This contact probability function depicts that when $r < r_c, f \approx 1$ but when $r > r_c, f \approx \left(r_c/r\right)^4$. The power-law decay with an exponent of 4 is consistent with the relationship between contact probability and spatial distances revealed in imaging studies (**Qi and Zhang, 2019**; **Wang et al., 2017**). The tanh function ensures the continuity of the function and its derivative around $r_c$ (**Appendix 1—figure 1**). Additionally, we truncated the ideal potential to be applicable for a sequence separation less than or equal to 100 MB and set the parameters for larger sequence separations to be zero. As shown in **Figure 4** of the main text, our parameterized ideal potential produced chromosomes with sizes comparable to imaging results. Incorporating longer-range interactions to improve the model further is straightforward but would also significantly increase the number of parameters.

Similar to the ideal potential discussed above, we have

$$U_{\text{compt}}\left(\mathbf{r}\right) = \sum_{i,j}\alpha_{\text{compt}}\left(T_i, T_j\right)f\left(r_{ij}\right), \tag{14}$$

where $T_i$ and $T_j$ denote the compartment types of beads $i$ and $j$ which can be $A$, $B$, or $C$. Therefore, CG beads of the same compartment types will share the same interaction parameter $\alpha_{compt}(T_i, T_j)$, which will be derived from average Hi-C contact frequencies as detailed in the following sections.

To account for specific interactions between chromosomes, we introduced the interchromosomal potential as

$$U_{inter}(\mathbf{r}) = \sum_{I,J>I} \sum_{i \in I, j \in J} \alpha_{inter}(I,J) f(r_{ij}).$$ (15)

$I, J \in \{1, 2, \ldots, 23\}$ index the haploid chromosomes, and parental and maternal chromosomes share identical parameters. This potential allows the model to capture interactions beyond those arising purely from compartmentalization as defined in *Equation 14*.

All parameters in the energy function are summarized in *Appendix 1—table 1*. The procedure used for parameter optimization is detailed in the following sections.

**Appendix 1—table 1.** Summary of the various terms of the chromosome energy function and the algorithms used for parameter optimization.
See also Appendix 1, section 'Hi-C inspired interactions for the diploid human genome' for detailed expression of the energy function and 'Adam optimizer for chromosome interaction parameters' for details on the optimization algorithm.

| Potentials | Functional forms | Parameter values |
|---|---|---|
| Bonding potential | $u_{bond}(r_{i,i+1})$ in *Equation 9* | Standard values in coarse-grained polymer models |
| Angular Potential | $u_{angle}(\vec{r}_{i,i+1}, \vec{r}_{i+1,i+2})$ in *Equation 9* | Standard values in coarse-grained polymer models |
| Soft-core potential | $u_{sc}(r_{ij})$ in *Equation 11* | Standard values in coarse-grained polymer models |
| Ideal potential | $U_{ideal}(\mathbf{r})$ in *Equation 12* | Values for $\alpha_{ideal}$ were obtained from optimizations against Hi-C data (see *Appendix 1—figure 3*). |
| Compartment potential | $U_{compt}(\mathbf{r})$ in *Equation 14* | Values for $\alpha_{compt}$ were obtained from optimizations against Hi-C data (see *Appendix 1—table 2*) |
| Inter potential | $U_{inter}(\mathbf{r})$ in *Equation 15* | Values for $\alpha_{inter}$ were obtained from optimizations against Hi-C data (see *Appendix 1—figure 4*) |

**Appendix 1—table 2.** Summary of interaction parameters between various compartment types, that is, $\alpha_{compt}$ defined in *Equation 14*.

| $\alpha_{AA}$ | −0.074185 |
|---|---|
| $\alpha_{AB}$ | 0.112285 |
| $\alpha_{AC}$ | 0.009947 |
| $\alpha_{BB}$ | 0.059981 |
| $\alpha_{BC}$ | 0.072481 |
| $\alpha_{CC}$ | 0.088825 |

## Nuclear landmark–nuclear landmark interactions

The general energy function for interactions among nuclear landmark particles is defined as

$$U_{NL} = U_{La}(\mathbf{r}) + U_{No}(\mathbf{r}) + U_{Sp-dP}(\mathbf{r}) + U_{EV}(\mathbf{r}).$$ (16)

The nuclear lamina was modeled as a particle mesh, and bonded potentials were introduced for nearest neighbor particles defined as

$$U_{La}(\mathbf{r}) = \sum_i \sum_{j \in n.n., j>i} K_2(r-r_o)^2 + K_3(r-r_o)^3 + K_4(r-r_o)^4, K_2 = K_3 = K_4 = 20\epsilon$$ (17)

with $r_o = 0.5\sigma$. $i$ indices all the lamina particles, and $j$ represents the nearest four neighbors around $i$ determined from the initial configuration for which the particles were placed on a Fibonacci grid. To avoid pairs $(i, j)$ being counted twice or more, we set $j$ always larger than $i$.

Short-ranged, non-bonded interactions were introduced among nuclear landmark particles to account for attractions that promote phase separation and the excluded volume effect. These interactions were modeled with a cut and shifted Lennard–Jones (LJ) potential defined as

$$U_{\mathrm{LJ}}\left(r_{ij}\right) = \begin{cases} 4\epsilon_{\mathrm{LJ}}\left(\left(\dfrac{\sigma_{\mathrm{LJ}}}{r_{ij}}\right)^{12} - \left(\dfrac{\sigma_{\mathrm{LJ}}}{r_{ij}}\right)^{6} - E_{\mathrm{cut}}\right), & \mathrm{for}\, r <= r_{\mathrm{cut}} \\[2ex] 0, & \mathrm{for}\, r > r_{\mathrm{cut}} \end{cases} \tag{18}$$

with $E_{\mathrm{cut}} = 4\epsilon\left(\left(\frac{\sigma}{r_{\mathrm{cut}}}\right)^{12} - \left(\frac{\sigma}{r_{\mathrm{cut}}}\right)^{6}\right)$. We note that when $r_{\mathrm{cut}}$ was set as $\sigma_{LJ} \times 2^{1/6}$, the potential has no attractive regime and only serves to prevent the overlap among particles, that is, the excluded volume effect.

For attractive interactions between nucleolus particles, and between type dP speckle particles, we set the parameters as $\epsilon_{\mathrm{LJ}} = 3.0, \sigma_{\mathrm{LJ}} = 0.5$, and $r_{\mathrm{cut}} = 1.5$. Therefore,

$$\begin{aligned} U_{\mathrm{No}}\left(\mathbf{r}\right) &= \sum_{j>i\in\mathrm{No}} U_{\mathrm{LJ}}\left(r_{ij}, \epsilon_{\mathrm{LJ}} = 3.0, \sigma_{\mathrm{LJ}} = 0.5, r_{\mathrm{cut}} = 1.5\right) \\ U_{\mathrm{Sp\text{-}dP}}\left(\mathbf{r}\right) &= \sum_{j>i\in\mathrm{Sp\text{-}dP}} U_{\mathrm{LJ}}\left(r_{ij}, \epsilon_{\mathrm{LJ}} = 3.0, \sigma_{\mathrm{LJ}} = 0.5, r_{\mathrm{cut}} = 1.5\right), \end{aligned} \tag{19}$$

where the sums iterate over pairs of nucleolus particles and speckle dP particles.

For the excluded volume effect between nucleolus and speckle particles, between dP and P particles, and between P particles, we set the parameters as $\epsilon_{\mathrm{LJ}} = 1.0, \sigma_{\mathrm{LJ}} = 0.5$, and $r_{\mathrm{cut}} = 0.5 \times 2^{1/6}$. These potentials are consistent with the estimated size of 0.5 $\sigma$ for speckle and nucleolus particles.

The excluded volume effect was also introduced between lamina and nucleolus particles and between the lamina and speckle particles to confine the nuclear bodies inside the nuclear envelope. We set the parameters as $\epsilon_{\mathrm{LJ}} = 1.0, \sigma_{\mathrm{LJ}} = 0.75$, and $r_{\mathrm{cut}} = 0.75 \times 2^{1/6}$. The value for $\sigma_{\mathrm{LJ}}$ was chosen based on a linear combination of the lamina particle size (1.0 $\sigma$) and the speckle/nucleolus particle size (0.5 $\sigma$).

Therefore, the excluded volume potential can be written as

$$\begin{aligned} U_{\mathrm{EV}}\left(\mathbf{r}\right) = &\sum_{i\in\mathrm{La}}\sum_{j\in\mathrm{No}} U_{\mathrm{LJ}}\left(r_{ij}, \epsilon_{\mathrm{LJ}} = 1.0, \sigma_{\mathrm{LJ}} = 0.75, r_{\mathrm{cut}} = 0.75 \times 2^{1/6}\right) \\ &+ \sum_{i\in\mathrm{La}}\sum_{j\in\mathrm{Sp}} U_{\mathrm{LJ}}\left(r_{ij}, \epsilon_{\mathrm{LJ}} = 1.0, \sigma_{\mathrm{LJ}} = 0.75, r_{\mathrm{cut}} = 0.75 \times 2^{1/6}\right) \\ &+ \sum_{i\in\mathrm{No}}\sum_{j\in\mathrm{Sp}} U_{\mathrm{LJ}}\left(r_{ij}, \epsilon_{\mathrm{LJ}} = 1.0, \sigma_{\mathrm{LJ}} = 0.5, r_{\mathrm{cut}} = 0.5 \times 2^{1/6}\right) \\ &+ \sum_{i\in\mathrm{Sp\text{-}P}}\sum_{j\in\mathrm{Sp\text{-}dP}} U_{\mathrm{LJ}}\left(r_{ij}, \epsilon_{\mathrm{LJ}} = 1.0, \sigma_{\mathrm{LJ}} = 0.5, r_{\mathrm{cut}} = 0.5 \times 2^{1/6}\right) \\ &+ \sum_{j>i\in\mathrm{Sp\text{-}dP}} U_{\mathrm{LJ}}\left(r_{ij}, \epsilon_{\mathrm{LJ}} = 1.0, \sigma_{\mathrm{LJ}} = 0.5, r_{\mathrm{cut}} = 0.5 \times 2^{1/6}\right) \\ &+ \sum_{j>i\in\mathrm{La}} U_{\mathrm{LJ}}\left(r_{ij}, \epsilon_{\mathrm{LJ}} = 1.0, \sigma_{\mathrm{LJ}} = 0.5, r_{\mathrm{cut}} = 0.5 \times 2^{1/6}\right). \end{aligned} \tag{20}$$

We used abbreviations to denote various nuclear landmarks, with La for the nuclear lamina, Sp-P for P-type speckle particles, Sp-dP for dP-type speckle particles, and No for nucleolus particles. All the interaction parameters for the nuclear landmarks are listed in **Appendix 1—table 3** for convenient reference.

**Appendix 1—table 3.** Summary of the interaction potentials among particles that make up the nuclear landmarks and their corresponding parameter values.
See also Appendix 1, section 'Nuclear landmark–nuclear landmark interactions' for further discussion and 'Unit conversion' for choosing the size of various particles, from which the $\sigma_{\mathrm{LJ}}$ were derived with a linear combination rule.

| Potentials | Function forms | Parameter values |
|---|---|---|
| Nucleolus/nucleolus | $U_{LJ}(r_{ij})$ in *Equation 18* | $\epsilon_{LJ} = 3.0$ and $r_{cut} = 1.5$ were chosen to mimic short-range attractions that produce an average of two nucleoli per cell. $\sigma_{LJ} = \sigma_{No} = 0.5$. |
| Sp-dP/Sp-dP (speckles) | $U_{LJ}(r_{ij})$ in *Equation 18* | $\epsilon_{LJ} = 3.0$ and $r_{cut} = 1.5$ were chosen to mimic short-range attractions that produce around 30 speckle droplets. $\sigma_{LJ} = \sigma_{Sp-dP} = 0.5$. |
| Sp-dP/Sp-P (speckles) | $U_{LJ}(r_{ij})$ in *Equation 18* | $\epsilon_{LJ} = 1.0$ and $r_{cut} = 0.5 \times 2^{1/6}$ were chosen as standard values to provide the excluded volume effect. $\sigma_{LJ} = \frac{\sigma_{Sp-dP}+\sigma_{Sp-P}}{2} = 0.5$. |
| Sp-P/Sp-P (speckles) | $U_{LJ}(r_{ij})$ in *Equation 18* | $\epsilon_{LJ} = 1.0$ and $r_{cut} = 0.5 \times 2^{1/6}$ were chosen as standard values to provide the excluded volume effect. $\sigma_{LJ} = \sigma_{Sp-P} = 0.5$. |
| Nucleolus/speckle | $U_{LJ}(r_{ij})$ in *Equation 18* | $\epsilon_{LJ} = 1.0$ and $r_{cut} = 0.5 \times 2^{1/6}$ were chosen as standard values to provide the excluded volume effect. $\sigma_{LJ} = \frac{\sigma_{No}+\sigma_{Sp}}{2} = 0.5$. |
| Nucleolus/lamina | $U_{LJ}(r_{ij})$ in *Equation 18* | $\epsilon_{LJ} = 1.0$ and $r_{cut} = 0.75 \times 2^{1/6}$ were chosen as standard values to provide the excluded volume effect. $\sigma_{LJ} = \frac{\sigma_{No}+\sigma_{La}^{*}}{2} = 0.75$. |
| Speckle/lamina | $U_{LJ}(r_{ij})$ in *Equation 18* | $\epsilon_{LJ} = 1.0$ and $r_{cut} = 0.75 \times 2^{1/6}$ were chosen as standard values to provide the excluded volume effect. $\sigma_{LJ} = \frac{\sigma_{Sp}+\sigma_{La}^{*}}{2} = 0.75$. |
| Lamina/lamina | $U_{LJ}(r_{ij})$ in *Equation 18* | $\epsilon_{LJ} = 1.0$ and $r_{cut} = 0.5 \times 2^{1/6}$ were chosen as standard values to provide the excluded volume effect. $\sigma_{LJ} = \sigma_{La} = 0.5$. |

* As mentioned in Appendix 1, section 'Mapping lamina bead size to real unit', a larger value for $\sigma_{La}$ was used here to provide a stronger excluded volume effect that prevents these particles from crossing the nucleus boundary or getting stuck in the space of the lamina particle mesh grid.

## Chromosome–nuclear landmark interactions

The energy function for interactions between chromosome and nuclear landmark particles is defined as

$$U_{GN} = U_{C\text{-}La}(\mathbf{r}) + U_{C\text{-}Sp}(\mathbf{r}) + U_{C\text{-}No}(\mathbf{r}). \qquad (21)$$

The functional form of the potential used to describe interactions between chromosomes and nuclear landmarks is inspired by experimental techniques that probe their contacts, such as Lamin B DamID and SON TSA-Seq. For example, the average contact probability between a chromatin bead $i$ and the nuclear lamina can be estimated as

$$p_i^{L} = \left\langle \sum_{j \in La} c(r_{ij}) \right\rangle, \qquad (22)$$

where $j$ indexes over the lamina particles. $c(r_{ij})$ is defined as

$$c(r_{ij}) = \frac{1}{2}\left(1 + \tanh\left[\eta(r_c - r_{ij})\right]\right). \qquad (23)$$

It is a switching function that approaches one for $r_{ij} < r_c$, a threshold distance at which we set chromatin and the lamina as in contact. We chose $\eta = 4.0$ to obtain a reasonable decay of contact probability between chromosomes and nuclear landmarks. $r_c = 0.75$ was selected as the average size of the lamina (1.0 $\sigma$) and chromatin (0.5 $\sigma$) particles.

For the computational model to reproduce the experimental contact probability, following the maximum entropy argument (*Roux and Weare, 2013*; *Zhang and Wolynes, 2015*), the interaction potential between chromosomes and the nuclear lamina adopts the following form:

$$U_{\text{C-La}}\left(\mathbf{r}\right) = \sum_{i \in \text{Chr}}\sum_{j \in \text{La}}\left\{ \frac{1}{2}\alpha_i^{\text{C-La}}\left(1 + \tanh\left[\eta\left(r_c - r_{ij}\right)\right]\right) \right. \\ \left. + U_{\text{LJ}}\left(r_{ij}, \epsilon_{\text{LJ}} = 1.0, \sigma_{\text{LJ}} = 0.75, r_{\text{cut}} = 0.75 \times 2^{1/6}\right) \right\}. \tag{24}$$

A similar argument to the one outlined above was used to derive the interactions among chromosomes from Hi-C data, that is, **Equations 12, 14, and 15** (**Hetzer, 2010**). The individual parameters $\alpha_i^{\text{C-La}}$ were optimized to ensure a match between simulated and experimental Lamin B DamID data. The second term was included to account for the excluded volume effect and prevent chromatin from moving outside the envelope.

The interaction potential between chromosomes and the speckles adopts a similar form defined as

$$U_{\text{C-Sp}}\left(\mathbf{r}\right) = \sum_{i \in \text{Chr}}\sum_{j \in \text{SP-dP}}\frac{1}{2}\alpha_i^{\text{C-Sp}}\left(1 + \tanh\left[\eta\left(r_c - r_{ij}\right)\right]\right). \tag{25}$$

The second sum for $j$ only includes dP-type speckle particles. The individual parameters $\alpha_i^{\text{C-Sp}}$ were optimized to ensure a match between simulated and experimental SON TSA-seq data.

Finally, the interaction potential between chromosomes and nucleoli is defined as

$$U_{\text{C-No}}\left(\mathbf{r}\right) = \sum_{i \in \text{Chr}}\sum_{j \in \text{No}}\frac{1}{2}\alpha_i^{\text{C-No}}\left(1 + \tanh\left[\eta\left(r_c - r_{ij}\right)\right]\right). \tag{26}$$

Because of the low data quality for the ChIP-Seq experiments for detecting chromatin-nucleoli contacts, we did not perform systematic optimizations for $\alpha_i^{\text{C-No}}$. Instead, we simply set them as $\alpha_i^{\text{C-No}} = P_i^{\text{N}}\epsilon$, with $\epsilon = 1.0$. $P_i^{\text{N}}$ is the probability for the chromatin bead $i$ to contact nucleoli as quantified by the software SPIN (**Wang et al., 2021**).

We list all the interaction parameters between chromosomes and the nuclear landmarks in **Appendix 1—table 4**.

**Appendix 1—table 4.** Summary of the interaction potentials between chromatin particles and nuclear landmarks and their corresponding parameter values.
See also Appendix 1, section 'Chromosome–nuclear landmark interactions' for further discussion and 'Adam optimizer for chromosome–nuclear body interaction parameters' for details on the optimization algorithm.

| Potentials | Functional forms | Parameter values |
|---|---|---|
| Chromatin-nucleolus | $U_{\text{C-No}}\left(r_{ij}\right)$ in **Equation 26** | $\eta = 4.0$ provides a smooth transition in the tanh function for contacts. $r_c = 0.75$ reflects the minimal distances between chromatin and nucleolus beads as reflected in the excluded volume potential defined in **Appendix 1—table 3**. The interaction strength of the $i$th chromatin bead $\alpha_i^{\text{C-No}} = P_i^{\text{N}}$, where $P_i^{\text{N}}$ is the probability for the chromatin bead $i$ to contact nucleoli as quantified by the software SPIN (**Wang et al., 2021**). |
| Chromatin-speckle | $U_{\text{C-Sp}}\left(r_{ij}\right)$ in **Equation 25** | $\eta = 4.0$ and $r_c = 0.75$ were similarly determined as in $U_{\text{C-No}}\left(r_{ij}\right)$. Value for the interaction strength of the $i$th chromatin bead $\alpha_i^{\text{C-No}}$ was obtained from optimizations against SON TSA-Seq data. |
| Chromatin-lamina | $U_{\text{C-La}}\left(r_{ij}\right)$ in **Equation 24** | $\eta = 4.0$ and $r_c = 0.75$ were similarly determined as in $U_{\text{C-No}}\left(r_{ij}\right)$. Value for the interaction strength of the $i$th chromatin bead $\alpha_i^{\text{C-No}}$ was obtained from optimizations against Lamin B DamID data. The extra Lennard Jones potential was included to provide the excluded volume effect, with $\epsilon_{\text{LJ}} = 1.0$ and $r_{\text{cut}} = 0.75 \times 2^{1/6}$ as standard values. $\sigma_{\text{LJ}} = \frac{\sigma_{\text{C}} + \sigma_{\text{La}}^*}{2} = 0.75$. |

* As mentioned in Appendix 1, section 'Mapping lamina bead size to real unit', a larger value for $\sigma_{\text{La}}$ was used here to provide a stronger excluded volume effect that prevents these particles from crossing the nucleus boundary or getting stuck in the space of the lamina particle mesh grid.

## Optimization of the whole nucleus model parameters

Below, we describe the procedures used to derive model parameters.

### Connecting imaging and Hi-C data with the contact function

The function $f(r)$ defined in **Equation 13** was used to determine the chromatin contact probabilities. The availability of spatial positions and Hi-C data makes possible the definition of a contact function, $f(r)$, that converts distances into contact probabilities. In particular, we determined $r_c$ as the value at which the simulated average interchromosomal contact probability $\langle f(r_c)\rangle^{\text{sim}}_{\text{inter}}$ matches the experimental value, that is,

$$\langle f(r)\rangle^{\text{sim}}_{\text{inter}} = f^{\text{exp}}_{\text{inter}}. \tag{27}$$

The angular brackets represent ensemble averaging, performed using the structures at 100 KB resolution reported in our previous work (**Kamat et al., 2023**). Matching simulation and experimental values produced $r_c = 0.54\sigma \approx 208$ nm. We note that this estimation for $r_c$ is comparable to the average bond length ($0.5\ \sigma$), thus ensuring that nearest neighbor genomic regions with contact probability close to 1, that is, $\langle f(r_{i,i+1})\rangle \approx 1$.

### Adam optimizer for chromosome interaction parameters

Mathematical expressions for the various energy terms in $U_{\text{Genome}}$ were designed such that their ensemble averages can be mapped onto combinations of contact frequencies measured in Hi-C. The correspondence between the energy functions and Hi-C measurements allows model parameterization with an efficient adaptive moment (Adam) algorithm (**Kingma and Ba, 2014**). Specifically, $\alpha_{\text{ideal}}(|i-j|)$, $\alpha_{\text{compt}}(T_i, T_j)$, and $\alpha_{\text{inter}}(I, J)$ were tuned to satisfy the following constraints:

$$\left\langle \sum_I \sum_{i,j\in I} f(r_{ij})\,\delta_{|i-j|,s} \right\rangle = \sum_I \sum_{i,j\in I} f^{\text{exp}}_{ij}\delta_{|i-j|,s}, \quad \text{for}\, s = 1, \cdots, n-1$$

$$\left\langle \sum_{i,j} f(r_{ij})\,\delta_{T_i,T_1}\delta_{T_j,T_2} \right\rangle = \sum_{i,j} f^{\text{exp}}_{ij}\delta_{T_i,T_1}\delta_{T_j,T_2}, \quad \text{for}\, T_1, T_2 \in \{A, B, C\}$$

$$\left\langle \sum_{i\in I, j\in J} f(r_{ij})\,\delta_{I,K_1}\delta_{J,K_2} \right\rangle \sum_{i\in I, j\in J} f^{\text{exp}}_{ij}\delta_{I,K_1}\delta_{J,K_2},$$

$$\text{for } K_1, K_2 \in \{1, \ldots, 23\} \tag{28}$$

where $\delta_{T_i,T_1}$ is the Kronecker delta function with the following definition:

$$\delta_{T_i,T_1} = \begin{cases} 1, & \text{if } T_i = T_1 \\ 0, & \text{otherwise} \end{cases} \tag{29}$$

The angular bracket represents the ensemble average, and $f^{\text{exp}}_{ij}$ is the corresponding experimental contact frequency.

During the optimization process, our aim was to minimize the disparity between experimental findings and simulated data. To achieve this, we defined the cost function as follows:

$$L = \sum_i \left(\langle f_i\rangle - f^{\text{exp}}_i\right)^2, \tag{30}$$

where the index $i$ iterates over all the constraints defined in **Equation 28**.

The details of the algorithm for parameter optimization are as follows:

1. Starting with a set of values for $\alpha_{\text{ideal}}(|i-j|)$, $\alpha_{\text{compt}}(T_i, T_j)$, and $\alpha_{\text{inter}}(I, J)$, we performed 50 independent 3-million-step long MD simulations to obtain an ensemble of nuclear configurations. The 500K steps of each trajectory are discarded as equilibration. We collected the configurations at every 2000 simulation steps from the rest of the simulation trajectories to compute the ensemble averages defined on the left-hand side of **Equationi 13**.

2. Check the convergence of the optimization by calculating the percentage of error defined as $\sum_i \left( \langle f_i \rangle - f_i^{\text{exp}} \right) / \sum_i f_i^{\text{exp}}$. The summation over $i$ includes all the average contact probabilities defined in **Equation 28**.

3. If the error is less than a tolerance value $e_{\text{tol}}$, the optimization has converged, and we stop the simulations. Otherwise, we update the parameters, $\alpha$, using the Adam optimizer (**Kingma and Ba, 2014**). With the new parameter values, we return to step one and restart the iteration.

## Adam optimizer for chromosome–nuclear body interaction parameters

Similar to those among chromatin particles, the interaction parameters between chromatin and nuclear landmarks were optimized with Adam's algorithm to reproduce experimental constraints.

The constraints that we aimed to reproduce were defined as follows:

$$
\begin{aligned}
\left\langle C_i^{\text{La}} \right\rangle &= \text{LAF}_i^{\text{exp}}, \ \text{for} \ i = 1, \cdots, N \\
\left\langle C_i^{\text{Sp}} \right\rangle &= \text{SAF}_i^{\text{exp}}, \ \text{for} \ i = 1, \cdots, N,
\end{aligned}
\tag{31}
$$

where $C_i^{\text{La}}$ and $C_i^{\text{Sp}}$ measure the contacts between chromatin bead $i$ and nuclear lamina and speckles, respectively, as defined in **Equations 42 and 45**. $\text{LAF}_i$ and $\text{LAF}_i$ denote the lamina and speckle association frequency for chromatin bead $i$ as measured in Lamin B DamID and SON TSA-Seq experiments. $N$ denotes the number of chromatin beads. We combined the constraints defined in **Equation 31** with those in **Equation 28** to simultaneously optimize the parameters using the iterative algorithm outlined in the previous section. We note that the interaction potential between chromatin and speckles defined in **Equation 25** did not use precisely the same function as in $C_i^{\text{Sp}}$. We chose to sum over all speckle dP particles, rather than identifying the droplets, which is difficult to do during the simulations.

## Parameter optimization for nuclear body–nuclear body interactions

As much remains to be known about the organization of nuclear bodies, we designed the interaction potentials and parameters based on qualitative observations without extensive fine-tuning. For example, we used the standard Lennard–Jones potential (**Equation 18**) to mimic short-range interactions. The lengthscales, $\sigma_{\text{LJ}}$, in these potentials, were chosen based on a linear combination of the size of interacting particles, as discussed in section 'Unit conversion'.

The interaction strength, $\epsilon_{\text{LJ}}$, was set as 1.0 to be on the same order as thermal energy ($k_{\text{B}} T$), when the potential was used to account for the excluded volume effect.

For attractive interactions that promote phase separation and nuclear body formation, we set $\epsilon_{\text{LJ}} = 3.0$. Smaller values failed to produce clustered nucleoli, while much larger values significantly decreased the fluidity of the resulting droplets. The same value was used for speckle dP particles and produced droplet numbers comparable to experimental observations (**Figure 2—figure supplement 1**).

## Unit conversion

The reduced unit for length scale is noted as $\sigma$. We set the nucleus radii as $13\sigma$. Assuming a nucleus with an average size of 5 μm, we have $\sigma = 385$ nm.

## Mapping chromatin bead size to real unit

We estimated the size of the chromosome bead as 192.5 nm based on super-resolution imaging data as follows. The median radius of gyration has been shown to follow a power-law scaling as a function of domain length with an exponent of 0.3 (**Boettiger et al., 2016**). Assuming that the radius of a domain is proportional to the radius of gyration, we have

$$
R \propto R_g \propto L^{0.3} \Rightarrow \frac{R_{1\text{MB}}}{R_{100KB}} = \left( \frac{1MB}{100KB} \right)^{0.3}.
\tag{32}
$$

We previously estimated the size of 1 MB bead as $R_{1\text{MB}} = \sigma = 385$ nm, and **Equation 32** yields the size of 100 KB as $R_{100\text{KB}} = 0.5\sigma$.

## Mapping lamina bead size to real unit

We chose the number and the diameter of lamina beads $N_{La}$, $\sigma_{La}$ by estimating the distance between nearest neighbor lamina beads. We found that at $N_{La} = 8000$, when the lamina particles were placed on the Fibonacci grid over the spherical surface, the average nearest neighbor distance was 0.52. Therefore, we set $\sigma_{La} = 0.5\sigma$ when considering the excluded volume effect between lamina particles. However, when modeling the excluded volume effect between lamina and chromatin, nucleolus, or speckle particles, we used $\sigma_{La} = 1.0$ (see *Equation 20*). A larger value provides a stronger excluded volume effect that prevents these particles from crossing the nucleus boundary or getting stuck in the space of the lamina particle mesh grid.

## Mapping nucleoli bead size to real unit

The size of nucleolus particles ($\sigma_{No}$) was estimated as follows. Since the average number of nucleoli inside a cell nucleus ranges from 2 to 5, we approximate the number of particles comprising individual droplets as $N_{No}/3$, assuming a total of three nucleoli. $N_{No}$ corresponds to the total number of nucleolus particles. With a space-filling model, the ratio of the volume between one nucleolus and the cell nucleus can be estimated as

$$\frac{(4\pi/3)\left(2^{1/6}\sigma_n/2\right)^3 (N_{No}/3)}{(4\pi/3)\,R_N^3} = \left(\frac{R_{No}}{R_N}\right)^3 \tag{33}$$

where $2^{1/6}\sigma_n/2$ denotes the effective radius of a nucleolus particle, and $R_N$ is the nucleus size. Using experimental values for the nucleolus and nucleus size (*Caragine et al., 2018*; *Caragine et al., 2019*) as $R_{No} = 0.5\mu m$ and $R_N = 5\mu m$, we have $\sigma_{No} = 0.5$.

## Mapping speckle bead size to real unit

A similar procedure as in the previous section was used to estimate the size of speckle particles $\sigma_{Sp}$. Since approximately 600 dP-type speckle particles form speckle clusters, each speckle cluster consists of around 20 particles. This estimation assumes a total of 30 speckle droplets in the system, consistent with the experimentally reported range of 20–50 speckles.

With a space-filling model, the ratio of the volume between one speckle and the cell nucleus can be estimated as

$$\frac{(4\pi/3)\left(2^{1/6}\sigma_{Sp}/2\right)^3 (N_{Sp})}{(4\pi/3)\,R_N^3} = \left(\frac{R_{Sp}}{R_N}\right)^3 \tag{34}$$

where $N_{Sp} = 20$. Using experimental values for the speckle and nucleus size (*Handwerger et al., 2005*) as $R_{Sp} = 0.3\mu m$ and $R_N = 5\mu m$, we have $\sigma_{Sp} = 0.5$.

## Mapping the reduced time unit to real time

We determined the timescale mapping by matching the simulated diffusion coefficient of chromatin particles with experimental values. The diffusion coefficient in our simulations can be estimated from the fluctuation-dissipation theorem (*Kubo, 1966*) as $D = \frac{k_B T}{\zeta}$, where the friction coefficient $\zeta = m\gamma$. Using the conversion from $\frac{k_B T}{m} = \frac{\sigma^2}{\tau_B^2}$, we have

$$D = \frac{k_B T}{\zeta} = \frac{k_B T}{m\gamma} = \frac{\sigma^2}{\tau_B^2 \gamma} = \frac{10^{-2}\sigma^2}{\tau_B}. \tag{35}$$

We used the simulation setup $\gamma^{-1} = 10^{-2}\tau_B$ when deriving the last equation.

In the meantime, from the Stokes–Einstein (SE) equation, we have $D = \frac{k_B T}{6\pi\eta r}$, where $\eta$ is the viscosity and $r = 0.25\sigma$ is the radius of chromatin beads. Therefore,

$$\frac{k_B T}{6\pi\eta r} = \frac{10^{-2}\sigma^2}{\tau_B}, \tag{36}$$

and

$$\tau_B = \frac{10^{-2}\sigma^2 \cdot 6\pi\eta r}{k_B T} = \frac{1.5 \times 10^{-2}\pi\eta\sigma^3}{k_B T}.$$

(37)

Setting the nucleoplasmic viscosity as $1 Pa \cdot s$ produces $\tau_B \approx 0.65 s$. This mapping produced diffusion coefficients and MSD curves that match well with experimental measurements presented in *Bronshtein et al., 2015*, as discussed in the main text. We note that the chosen value for the nucleoplasmic viscosity indeed falls into the range of reported experimental values from $10^{-1} Pa \cdot s$ to $10^2 Pa \cdot s$ (*Platani et al., 2002*; *Tseng et al., 2004*).

## Molecular dynamics simulation details

### Initial configurations for simulations

Due to the slow relaxation dynamics of whole chromosomes relative to the simulation timescale, the reported results are sensitive to the configurations used to initialize the simulations. Therefore, we designed the following protocol to prepare the initial configurations and ensure the biological relevance of simulation results.

We first created a total of 1000 configurations for the genome by sequentially generating the conformation of each one of the 46 chromosomes as follows. For a given chromosome, we start by placing the first bead at the center (origin) of the nucleus. The positions of the following beads, $i$, were determined from the $(i-1)$-th bead as $r_i = r_{i-1} + 0.5v$. **v** is a normalized random vector, and 0.5 was selected as the bond length between neighboring beads. To produce globular chromosome conformations, we rejected vectors, **v**, that led to bead positions with distance from the center larger than $4\sigma$. Upon creating the conformation of a chromosome $i$, we shift its center of mass to a value $r_{com}^i$ determined as follows. We first compute a mean radial distance, $r_o^i$ with the following equation:

$$\frac{6\sigma - r_o^i}{r_o^i - 2\sigma} = \frac{D_{hi} - D}{D - D_{lo}},$$

(38)

where $D_i$ is the average value of Lamin B DamID profile for chromosome $i$. $D_{hi}$ and $D_{lo}$ represent the highest and lowest average DamID values of all chromosomes, and $6\sigma$ and $2\sigma$ represent the upper and lower bound in radial positions for chromosomes. As shown in *Appendix 1—figure 2*, the average Lamin B DamID profiles are highly correlated with normalized chromosome radial positions as reported by DNA MERFISH (*Su et al., 2020*), supporting their use as a proxy for estimating normalized chromosome radial positions. We then select $r_{com}^i$ as a uniformly distributed random variable within the range $\left[r_o^i - 2\sigma, r_o^i + 2\sigma\right]$. Without loss of generality, we randomly chose the directions for shifting all 46 chromosomes.

We further relaxed the 1000 configurations to build more realistic genome structures. Following an energy minimization process, 1-million-step MD simulations were performed starting from each configuration. Simulations were performed with the following energy function:

$$U_{Relax} = U_{Genome} + U_{C-La}^{EV},$$

(39)

where $U_{Genome}$ is defined as in *Equation 7*. $U_{G-La}$ is the excluded volume potential between chromosomes and lamina, that is, only the second term in *Equation 24*. Parameters in $U_{Genome}$ were from a preliminary optimization. The end configurations of the MD simulations were collected to build the final configuration ensemble (FCE).

We further computed the Pearson correlation coefficient of pairwise interchromosomal contacts between different structures in FCE (see section 'Computing pairwise interchromosomal contact probabilities'). As shown in *Figure 6—figure supplement 2A*, the probability distribution of these correlation coefficients is comparable with that determined from DNA-MERFISH structures, supporting the biological relevance of the structural diversity in the constructed ensemble.

From 1000 relaxed configurations, we selected a subset of structures to initialize simulations presented in the main text. An optimization procedure was introduced for structure selection. We start this procedure by randomly select $N$ structures to build the initial configuration ensemble (ICE). We then iteratively go through every configuration in ICE and replace with a structure from FCE that's not already included in ICE. We then compute the Pearson correlation coefficient between new average ICE interchromosomal contact probabilities and experimental values. If the Pearson correlation coefficient is higher than the value determined from the original ICE, the new structure

is accepted and the ICE is updated. Otherwise, the new structure is rejected. We stop the selection process for when the Pearson correlation coefficient stops improving.

We found that as $N$ increases, the agreement between ICE interchromosomal contact probabilities and experimental values continue to increase (*Figure 6—figure supplement 2B*). We set $N = 50$, which produces a Pearson correlation coefficient between ICE and experimental interchromosomal contact probabilities of 0.9. Further increasing $N$ does not significantly improve the agreement but incurs more computational cost.

It is worth noting that the outcomes of the selection procedure depend on the initial set of configurations included in ICE at the beginning. However, we found that the ICEs produced from 20 independent trials are highly correlated (*Figure 6—figure supplement 2C*) and all reproduce the heterogeneity in interchromosomal contacts seen in DNA MERFISH data (*Figure 6—figure supplement 2D*). Therefore, the selection procedure is robust and can produce biologically meaningful configurations to initialize simulations.

With the chromosome positions prepared, we randomly placed 300 nucleoli and 1600 speckle particles inside the nucleus to complete the set up of initial configurations.

## Langevin dynamics simulations

We used the Langevin integrator with the damping coefficient $\gamma^{-1} = 10$ to control the temperature at $T = 1.0$ for simulations used for parameter optimization and for producing an ensemble of nucleus structures. Langevin dynamics simulations allow faster chromosome movements, compared to Brownian dynamics simulations, facilitating the conformational sampling. In these simulations, the lamina particles were frozen and no explicit dynamics were considered for the nuclear envelope.

## Brownian dynamics simulations

We also performed Brownian dynamics simulations with damping coefficient $\gamma^{-1} = 10^{-2}$ to control the temperature at $T = 1.0$. These simulations provide better approximations of the overdamped dynamics of chromatin for direct comparison with live cell imaging studies. As detailed in section 'Unit conversion', upon mapping the coarse-grained timescale to the physical unit, Brownian dynamics simulations produce diffusion coefficients for telomeres comparable to experimental values (see *Figure 5*).

## Nuclear envelope deformation simulations

We performed Langevin dynamics simulations to investigate the impact of nuclear envelope deformation on genome organization. To induce a compressing force along the $z$-axis, we introduced a harmonic potential in the form of

$$U_{\text{compress}} = \sum_{i=1}^{N_{\text{La}}} k \times \frac{z_i^2}{R_N}. \tag{40}$$

where $z_i$ is the $z$ coordinate of the $i$th lamina bead, and $N_{\text{La}}$ represents the total number of lamina beads. The particles in the system evolve under the combined effect of $U_{\text{compress}}$ and $U_{\text{Nucleus}}$ defined in *Equation 6*.

## Details of simulation data analysis

The computer simulations yield 3D coordinates of the diploid genome. However, when comparing directly with experimental data processed for the haploid genome, unless stated otherwise, we computed averages across paternal and maternal chromosomes to ascertain various genome-wide properties as listed below.

## Computing simulated contact probabilities

Simulated contact probability maps were computed by averaging over chromosome configurations collected from all trajectories. For a given configuration, the contact probability between two chromatin segments ($i$ and $j$) was evaluated using the contact function defined in *Equation 13*.

## Computing the Pearson correlation coefficients between experimental and simulated contact maps

We computed the Pearson correlation coefficients (PCCs) between experimental and simulated contact maps in *Figure 4A* and *Figure 4—figure supplement 1* as

$$r = \frac{\sum_{i=1}^{n} (x_i - \bar{x})(y_i - \bar{y})}{\sqrt{\sum_{i=1}^{n} (x_i - \bar{x})^2} \sqrt{\sum_{i=1}^{n} (y_i - \bar{y})^2}}. \tag{41}$$

where $x_i$ and $y_i$ represent the experimental and simulated contact probabilities, and $n$ is the total number of data points. Only non-redundant data points, that is, half of the pairwise contacts, are used in the PCC calculation.

## Computing pairwise interchromosomal contact probabilities

For a given genome structure, we computed the pairwise interchromosomal contacts as follows. For every pair of chromosomes, we determined their contact probability by averaging all genomic pairs from two chromosomes using *Equation 13*. We then averaged over all four pairs of diploid chromosomes to compute the haploid average contacts. In total, there are $C_{22}^2 = 231$ contact pairs between haploid chromosomes excluding the sex chromosomes.

## Distances from nuclear bodies and association frequencies

The contacts of a chromatin bead $i$ with the nuclear lamina were evaluated as

$$C_i^{\text{La}} = \frac{1}{N_t} \sum_t \sum_{j \in \text{La}} \frac{1}{2} \left( 1 + tanh \left[ \eta \left( r_c - r_{i,j} \right) \right] \right) \tag{42}$$

with $r_c = 0.75\sigma$. We average over the ensemble of nuclear configurations and homologs to compute the in silico Lamin B DamID signal as

$$\text{DamID}_i = \log_2 \left( \frac{\left\langle C_i^{\text{La}} \right\rangle}{\bar{C}^{\text{La}}} \right), \tag{43}$$

where the angular brackets indicate ensemble averaging. $\bar{C}^{\text{La}}$ is defined as the genome wide average of $\left\langle C_i^{\text{La}} \right\rangle$.

For chromatin-speckle contacts, we first identified the speckles formed at any given structure using the density-based spatial clustering algorithm DBSCAN (*Ester et al., 1996*) as implemented in the scikit library for Python (*Pedregosa et al., 2011*). For the identified droplets, we computed their center of mass coordinates, $\vec{r}^{com}$ and the radius of gyration, $R$. With the identified clusters, we then determined the distance from the $i$th chromatin bead to the $s$th speckle as

$$d_{i,s} = \| \vec{r}_i - \vec{r}_s^{com} \| - R_s, \tag{44}$$

where $\| \cdot \|$ represents the L2 norm. We subtract the radius of the speckle cluster in the above equation to determine the distance to the droplet surface. From the list of distances to different speckles, the contact between chromatin bead $i$ and speckles is computed as

$$C_i^{\text{Sp}} = \frac{1}{N_s} \sum_s \frac{1}{2} \left( 1 + tanh \left[ \eta \left( d_c - d_{i,s} \right) \right] \right), \tag{45}$$

where we sum over all the $N_s$ speckle clusters. A similar expression was used for determining the contacts between chromatin and nucleoli.

Finally, we average over the ensemble of nuclear configurations and homologs to compute the in silico SON TSA-Seq signal as

$$TSA_i = \log_2 \left( \frac{\left\langle C_i^{Sp} \right\rangle}{\bar{C}} \right),$$ (46)

where the angular brackets indicate ensemble averaging. $\bar{C}^{Sp}$ is defined as the genome wide average of $\left\langle C_i^{Sp} \right\rangle$.

## Computing simulated normalized chromosome radial positions

For a given chromosome $i$, we first determined its center of mass position denoted as $C_i$. Starting from the center of the nucleus, $O$, we extend the vector $v_{OC}$ to identify the intersection point with the nuclear lamina as $P_i$. The normalized radial position of chromosome $i$ is then defined as $\frac{\|v_{OC_i}\|}{\|v_{OP_i}\|}$, where $\|.\|$ represents the L2 norm.

## Computing simulated chromosome radii of gyration

The radius of gyration for a chromosome is computed as

$$R_g = \sqrt{\frac{\sum_i^n \| r_i - r_{com}\|}{n}},$$ (47)

where $r_{com}$ and $n$ are the center of mass and the number of beads of the chromosome. $i$ indices over all the chromosome beads and $r_i$ correspond to the Cartesian coordinates of bead $i$. $\|.\|$ represents the L2 norm.

## Computing simulated mean-square displacement

MSD for telomeres were computed as

$$\left\langle r^2 \left( \Delta t \right) \right\rangle = \frac{1}{N_{traj}} \sum_{t=1}^{N_{traj}} \frac{1}{N_{step}} \sum_{i=1}^{N_{step}} \left[ r^t \left( (i-1)\,\delta t + \Delta t \right) - r^t \left( (i-1)\,\delta t \right) \right]^2,$$ (48)

where $\Delta t, \delta t$, and $N_{step}$ represent the time interval, the time step, and the total number of steps, respectively. The summation over $t$ corresponds to averaging over eight independent trajectories. MSDs telomeres from paternal and maternal chromosomes are separately computed and analyzed.

## Details of experimental data analysis

### Interchromosomal contacts from DNA MERFISH data

We collected the DNA MERFISH data reported in *Su et al., 2020* to construct the experimental ensemble of 5455 genome structures. For each structure, we computed the pairwise interchromosomal contacts following the procedure outlined in section 'Computing pairwise interchromosomal contact probabilities'.

To better visualize and analyze interchromosomal contacts, we applied the Uniform Manifold Approximation and Projection (UMAP) technique as implemented in software package umap-learn (*McInnes et al., 2018*; *Moshtagh, 2005*), with default parameters to reduce the 231 haploid contacts into two dimensions. All 5455 DNA MERFISH structures were included in this analysis.

The same transformations produced from the UMAP analysis of experimental structures were applied to in silico configurations to produce results shown in *Figure 6—figure supplement 2C and D*.

### Computing experimental normalized chromosome radial positions

We followed the same procedure outlined in section 'Computing simulated normalized chromosome radial positions' to compute the experimental values. To determine the center of the nucleus using DNA MERFISH data, we used the algorithm, minimum volume enclosing ellipsoid (MVEE) (*Moshtagh, 2005*), to fit an ellipsoid for each genome structure. The optimal ellipsoid defined as $(x-c)^T A (x-c) \equiv 1$ is obtained by optimizing $\min \left( \log \left( \det [\mathbf{A}] \right) \right)$ subjecting to the constraint that $(x_i - c)^T A (x_i - c) \leq 1$. $x_i$ correspond to the list of chromatin positions determined experimentally.

## Computing experimental radii of gyration

We computed the experimental radii of gyration with using the same expression as that for analyzing simulated structures (*Equation 47*).

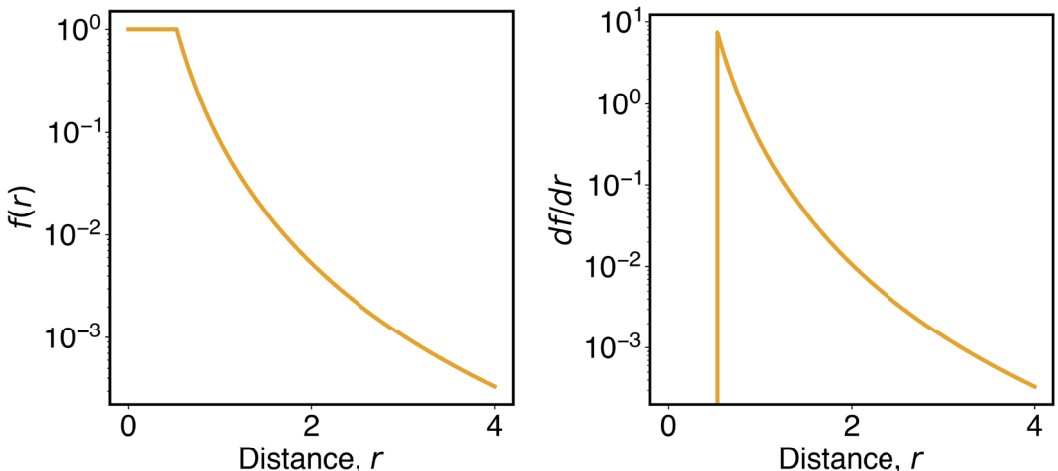

**Appendix 1—figure 1.** The function defined in *Equation 13* smoothly switches from high to low contact probabilities. The left and right panels plot the function and its derivative as a function of the distance, *r*.

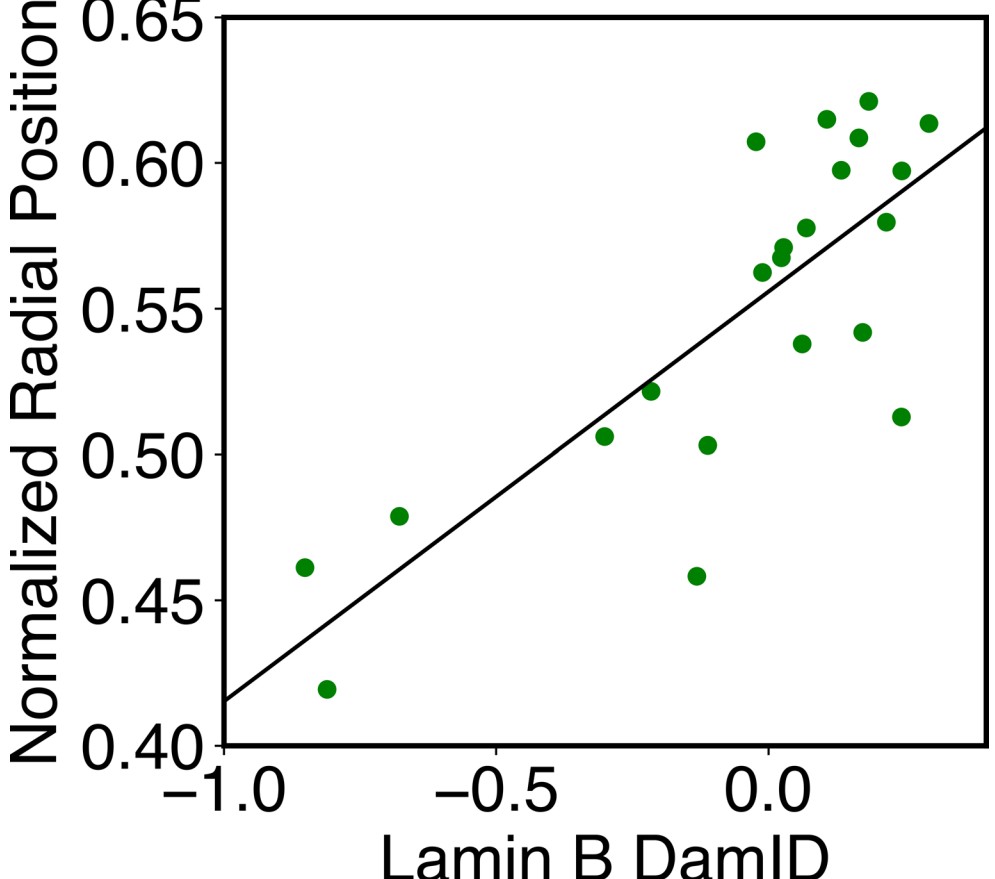

**Appendix 1—figure 2.** Correlation between average DamID profiles of individual chromosomes with their normalized radial positions. The normalized radial positions were determined using the average value of all cells reported from DNA MERFISH data (*Su et al., 2020*). The correlation coefficient between the two datasets is 0.8.

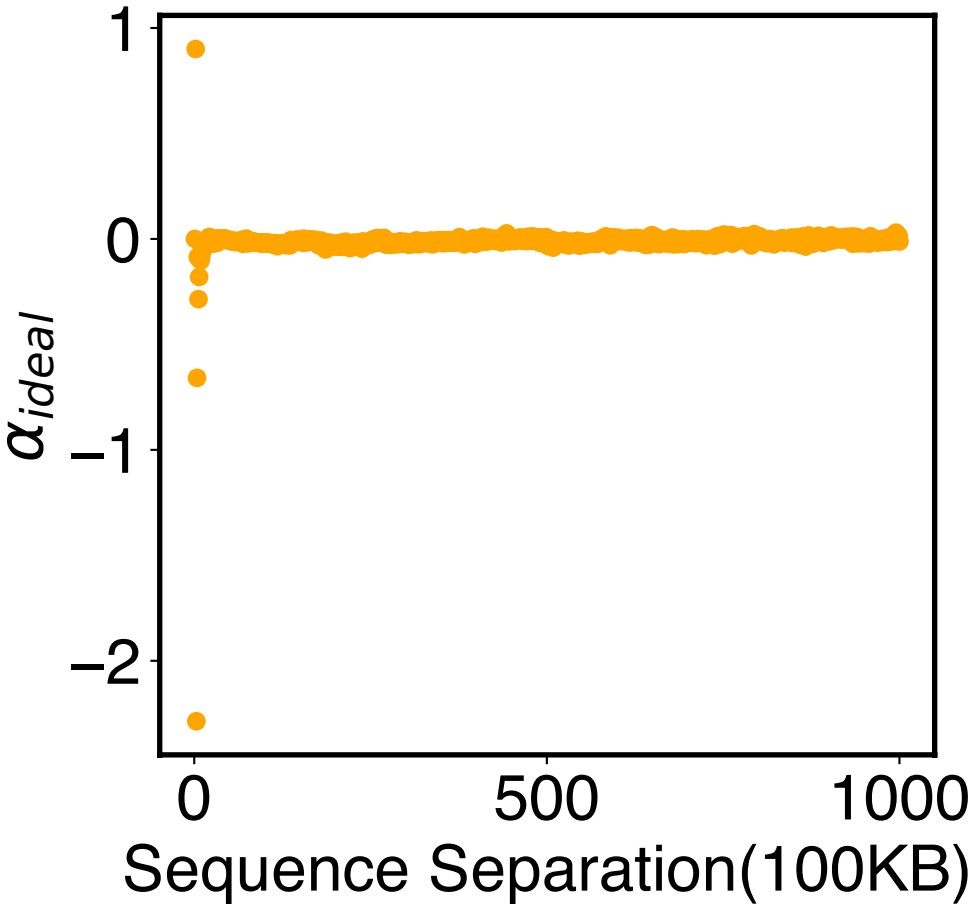

**Appendix 1—figure 3.** Parameters of the ideal potential, $\alpha_{\text{ideal}}$, as defined in *Equation 12*. Numerical values for $\alpha_{\text{ideal}}$ are included in the software's GitHub repository.

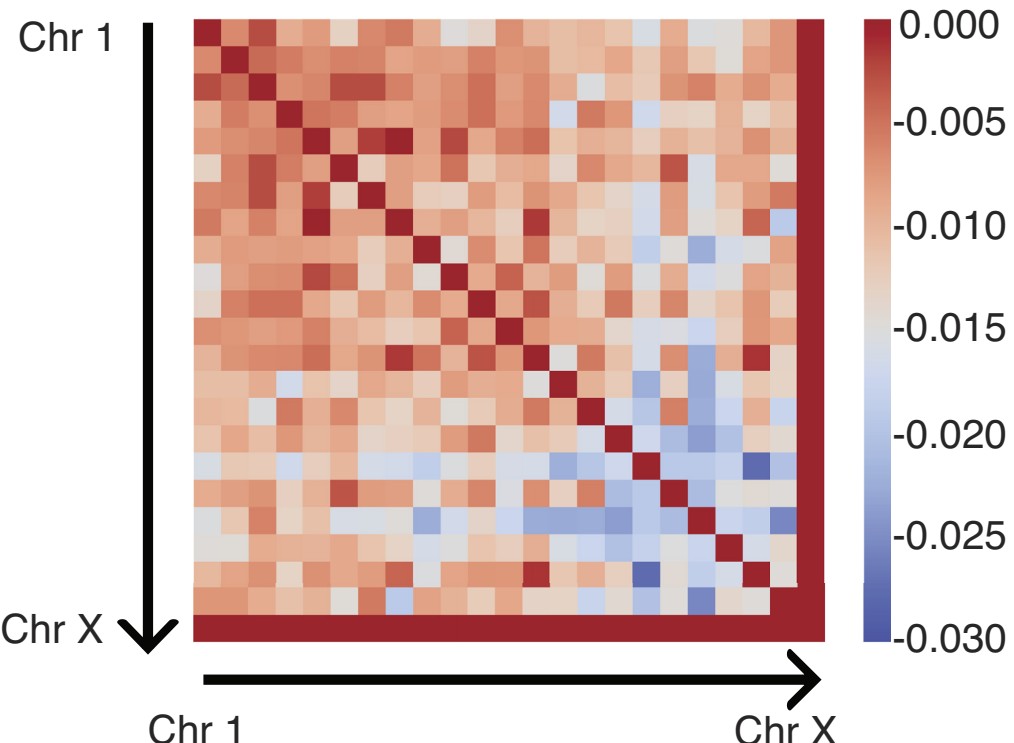

**Appendix 1—figure 4.** Parameters of the inter potential, $\alpha_{\text{inter}}$, as defined in *Equation 15*. Numerical values for $\alpha_{\text{inter}}$ are included in the software's GitHub repository.

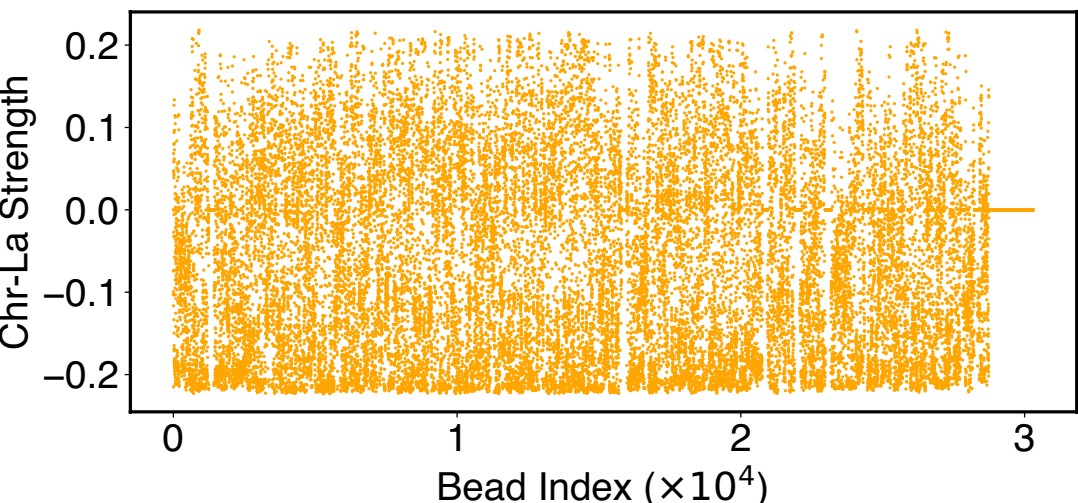

**Appendix 1—figure 5.** Parameters of the chromosome-lamina potential, $\alpha^{C-La}$, as defined in *Equation 24*. Numerical values for $\alpha^{C-La}$ are included in the software's GitHub repository.

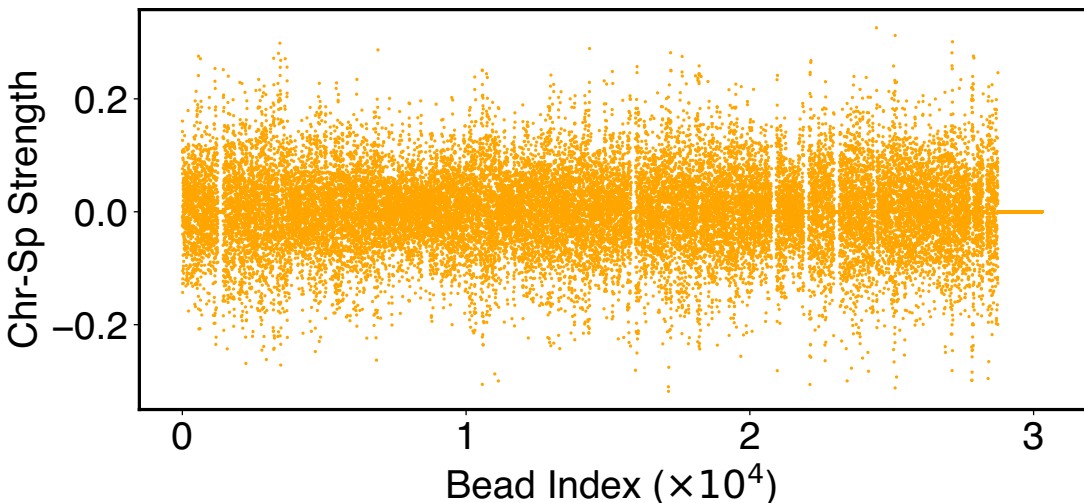

**Appendix 1—figure 6.** Parameters of the chromosome-lamina potential, $\alpha^{C-Sp}$, as defined in **Equation 25**. Numerical values for $\alpha^{C-Sp}$ are included in the software's GitHub repository.

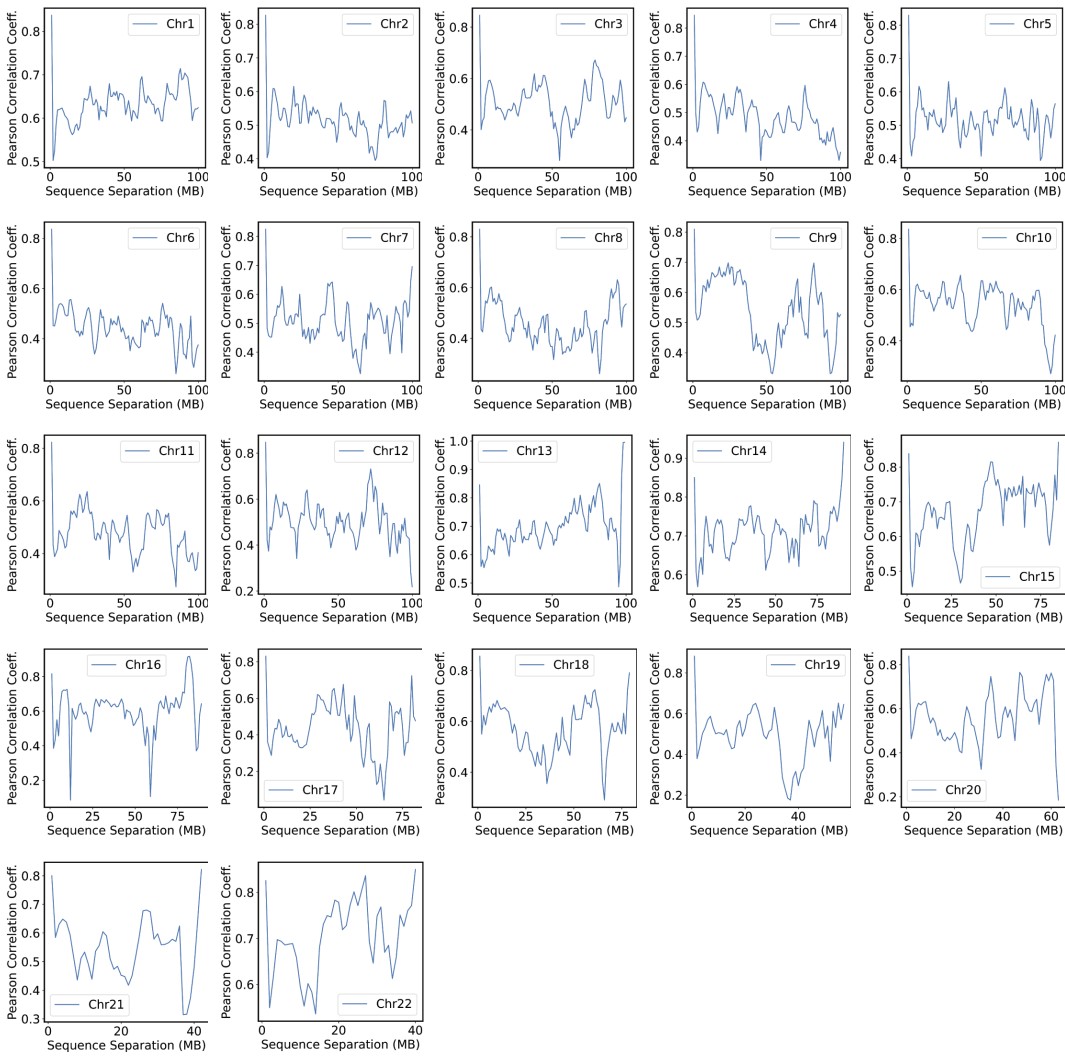

**Appendix 1—figure 7.** Pearson correlation coefficients between experimental and simulated contact probabilities at various sequence separations within specific chromosomes. For each chromosome, we first gathered a set of experimental contacts alongside a matching set of simulated ones for genomic pairs within a particular separation range. The Pearson correlation coefficient at the corresponding sequence separation was then determined using *Equation 41*. We limited the calculations to half of the chromosome length to ensure the availability of sufficient data.

