## [Editor Report · eLife assessment]

This **important** work significantly advances the field of computational modeling of genome organization through the development of OpenNucleome. The evidence supporting the tool's effectiveness is **compelling** as the authors compare their predictions with experimental data. It is anticipated that OpenNucleome will attract significant interest from the biophysics and genomics communities.

---

## [Referee Report · Reviewer #1 (Public Review)]

Summary:

In this paper the authors develop a comprehensive program to investigate the organization of chromosome structures at 100 kb resolution. It is extremely well executed. The authors have thought through all aspects of the problem. The resulting software will be most useful to the community. Interestingly they capture many experimental observations accurately. I have very little complaints.

Strengths:

A lot of details are provided. The success of the method is well illustrated. Software is easily available,

Weaknesses:

The number of parameters in the energy function is very large. Any justification? Could they simply be the functions?

What would the modification be if the resolution is increased?

They should state that the extracted physical values are scale dependent. Example, viscosity.

---

## [Referee Report · Reviewer #2 (Public Review)]

Summary:

In this work, Lao et al. develop an open-source software (OpenNucleome) for GPU-accelerated molecular dynamics simulation of the human nucleus accounting for chromatin, nucleoli, nuclear speckles, etc. Using this, the authors investigate the steady-state organization and dynamics of many of the nuclear components.

Strengths:

This is a comprehensive open-source tool to study several aspects of the nucleus, including chromatin organization, interactions with lamins and organization, and interactions with nuclear speckles and nucleoli. The model is built carefully, accounting for several important factors and optimizing the parameters iteratively to achieve experimentally known results. Authors have simulated the entire genome at 100kb resolution (which is a very good resolution to simulate and study the entire diploid genome) and predict several static quantities such as the radius of gyration and radial positions of all chromosomes, and time-dependent quantities like the mean-square displacement of important genomic regions.

Weaknesses:

One weakness of the model is that it has several parameters. Some of them are constrained by the experiments. However, the role of every parameter is not clear in the manuscript.

---

## [Referee Report · Reviewer #3 (Public Review)]

Summary:

The authors present OpenNucleome, a computational tool for simulating the structure and dynamics of the human nucleus. The software models nuclear components, including chromosomes and nuclear bodies, and incorporates GPU acceleration for potential performance gains. The authors aim to advance the understanding of nuclear organization by providing a tool that aligns with experimental data and is accessible to the genome architecture research community.

Strengths:

OpenNucleome provides a model of the nucleus, contributing to the advancement of computational biology.

Utilizing GPU acceleration with OpenMM may offer potential performance improvements.

Weaknesses:

It could still take advantage of clearer explanations regarding the generation and usage of input and output files and compatibility with other tools.

---

## [Author Response]

The following is the authors’ response to the original reviews.

**Reviewer 1:**
Comment 0: In this paper, the authors develop a comprehensive program to investigate the organization of chromosome structures at 100 kb resolution. It is extremely well executed. The authors have thought through all aspects of the problem. The resulting software will be most useful to the community. Interestingly they capture many experimental observations accurately.I have very few complaints.

We appreciate the reviewer’s strong assessment of the paper’s significance, novelty, and broad interest, and we thank them for the detailed suggestions and comments.

Comment 1: The number of parameters in the energy function is very large. Is there any justification for this? Could they simplify the functions?

We extend our gratitude to the reviewer for their insightful remarks. The parameters within our model can be categorized into two groups: those governing chromosome-chromosome interactions and those governing chromosome-nuclear landmark interactions.

In terms of chromosome-chromosome interactions, the parameter count is relatively modest compared to the vast amount of Hi-C data available. For instance, while the whole-genome Hi-C matrix at the 100KB resolution encompasses approximately 303212 contacts, our model comprises merely six parameters for interactions among different compartments, along with 1000 parameters for the ideal potential. As outlined in the supporting information, the ideal potential is contingent upon sequence separation, with 1000 chosen to encompass bead separations of up to 100MB. While it is theoretically plausible to reduce the number of parameters by assuming interactions cease beyond a certain sequence separation, determining this scale a priori presents a challenge.

During the parameterization process, we observed that interchromosomal contacts predicted solely based on compartmental interactions inadequately mirrored Hi-C data. Consequently, we introduced 231 additional parameters to more accurately capture interactions between distinct pairs of autosomes. These interactions may stem from factors such as non-coding RNA or proteins not explicable by simple, non-specific compartmental interactions.

Regarding parameters concerning chromosome-nuclear landmark interactions, we have 30321 parameters for speckles and 30321 for the nuclear lamina. To streamline the model, we opted to assign a unique parameter to each chromatin bead. However, it is conceivable that many chromatin beads share a similar mechanism for interacting with nuclear lamina or speckles, potentially allowing for a common parameter assignment. Nonetheless, implementing such simplification necessitates a deeper mechanistic understanding of chromosome-nuclear landmark interactions, an aspect currently lacking.

As our comprehension of nuclear organization progresses, the interpretability of parameter counts may improve, facilitating their reduction.

Comment 2: What would the modification be if the resolution is increased?

To increase the resolution of chromatin, we can in principle keep the same energy function as defined in Eq. 6. In this case, we only need to carry out further parameter optimization.

However, transitioning to higher resolutions may unveil additional features not readily apparent at 100kb. Notably, chromatin loops with an average size of 200kb or smaller have been identified in high-resolution Hi-C data [1]. To effectively capture these loops, new terms in the energy function must be incorporated. For instance, Qi and Zhang [2] employed additional contact potentials between CTCF sites to account for loop formation. Alternatively, an explicit loop-extrusion process could be introduced to model loop formation more accurately.

Comment 3: They should state that the extracted physical values are scale-dependent. For example, viscosity.

We thank the reviewer for the comment and would like to clarify that our model does not predict the viscosity. The nucleoplasmic viscosity was set as 1Pa · s to produce a diffusion coefficient that reproduces experimental value. The exact value for the nucleoplasmic viscosity is still rather controversial, and our selected value falls in the range of reported experimental values from 10−1Pa·s to 102Pa · s.

We have modified the main text to clarify the calculation of the diffusion coefficient.

“The exponent and the diffusion coefficient Dα = (27±11)×10−4μm2 · s−α both match well with the experimental values [cite], upon setting the nucleoplasmic viscosity as 1Pa · s (see Supporting Information Section: Mapping the reduced time unit to real time for more details).”

Reviewer 2:Comment 0: In this work, Lao et al. develop an open-source software (OpenNucleome) for GPU-accelerated molecular dynamics simulation of the human nucleus accounting for chromatin, nucleoli, nuclear speckles, etc. Using this, the authors investigate the steady-state organization and dynamics of many of the nuclear components.

We thank the reviewer for summary of our work.

Comment 1: The authors could introduce a table having every parameter and the optimal parameter value used. This would greatly help the reader.

We would like to point out that model parameters are indeed provided in Table 1, 2, 3, 4, and Appendix 1 - figure 3. In these tables, we further provided details on how the parameters were determined.

Given the large number of parameters for the ideal potential (1000), we opted to plot it rather than listing out all the numbers. We added three new figures to plot the interaction parameters between chromosomes, between chromosomes and speckles, and between chromosomes and the nuclear lamina. Numerical values can be found online in the GitHub repository (parameters).

Comment 2: How many total beads are simulated? Do all beads have the same size?

The total number of the coarse-grained beads is 70542, including 60642 chromatin beads, 300 nucleolus beads, 1600 speckle beads, and 8000 nuclear lamina beads. The radius of the chromatin, nucleolus, and speckle beads is 0.25, while that of the lamina bead is 0.5. More information of the size and number of the beads are discussed in the Section: Components of the whole nucleus model.

Comment 3: In 17, what is the 3rd and 4th powers mean? What necessitates it?

The potential defined in Equation 17 follows the definition of class2 bond in the LAMMPS package (LAMMPS docs). Compared to a typical harmonic potential, the presence of higher order terms produces sharper increase in the energy at large distances (Author response image 1). This essentially reduces the flucatuation of bond length in simulations.

**Author response image 1. sa4fig1:** Comparison between the Class2 potential (defined in Eq. 17) and the Harmonic potential (K(r − r0)2, with K = 20 and r0 = 0.5).

Comment 4: What do the X-axis and Y-axis numbers in Figure 5A and 5B mean? What are their units?

We apologize for the lack of clarify in our original figure. In Fig. 5A, the X and Y axis depicts the simulated and experimental radius of gyration (Rg) for individual chromosomes, as indicated in the title of the figure. Similarly, in Fig. 5B, the X and Y axis depicts the simulated and experimental radial position of individual chromosomes.

We have converted the chromosome Rg values into reduced units and labeled the corresponding axes in the updated figure (Fig. 5). The normalized radial position is unitless and its detailed definition is included in the supporting information Section: Computing simulated normalized chromosome radial positions. We updated the figure caption to provide an explicit reference to the SI text.

**Reviewer 3:**
Comment 0: In this work, the authors present the development of OpenNucleome, a software for simulating the structure and dynamics of the human nucleus. It provides a detailed model of nuclear components such as chromosomes and nuclear bodies, and uses GPU acceleration for better performance based on the OpenMM package. The work also shows the model’s accuracy in comparisons with experimental data and highlights the utility in the understanding of nuclear organization. While I consider this work a good tool for the genome architecture scientific community, I have some comments and questions that could further clarify the usage of this tool and help potential users. I also have a few questions that would help to clarify the technique and results and some suggestions for references.

We appreciate the reviewer’s strong assessment of the paper’s significance, novelty, and broad interest, and we thank them for the detailed suggestions and comments.

Comment 1: Could the authors elaborate on what they consider to be ’well-established and easily adoptable modeling tools’?

By well established, we meant that models that have been extensively validated and verified, and are highly regarded by the community.

By easily adoptable, we meant that tools that are well documented and can be relatively easily learned by new groups without help from the developers.

We have revised the text to clarify our meaning.

“Despite the progress made in computational modeling, the absence of well-documented software with easy-to-follow tutorials pose a challenge.”

Comment 2: Recognizing the value of a diverse range of tools in the community, the Open-MiChroM tool is also an open-source platform built on top of OpenMM. The documentation shows various modeling approaches and many tutorials that contain different approaches besides the MiChroM energy function. How does OpenNucleome compare in terms of facilitating crossvalidation and user accessibility? The two tools seem to be complementary, which is a gain to the field. I recommend adding one or two sentences in the matter. Also, while navigating the OpenNucleome GitHub, I have not found the tutorials mentioned in the text. I also consider a barrier in the process of generating necessary input files. I would suggest expanding the tutorials and documentation to help potential users.

We thank the reviewer for the excellent comments. We agree that while many of the tutorials were included in the original package, they were not as clearly documented. We have revised them extensively to to now present:

• A tutorial for optimizing chromosome chromosome interactions.

• A tutorial for optimizing chromosome nuclear landmark interactions.

• A tutorial for building initial configurations.

• A tutorial for relaxing the initial configurations.

• A tutorial for selecting the initial configurations.

• A tutorial for setting up performing Langevin dynamics simulations.

• A tutorial for setting up performing Brownian dynamics simulations.

• A tutorial for setting up performing simulations with deformed nucleus.

• A tutorial for analyzing simulation trajectories.

• A tutorial for introducing new features to the model.

These tutorials and our well-documented and open source code (https://zhanggroup-mitchemistry.github.io/OpenNucleome) should significantly promote user accessibility. Our inclusion of python scripts for analyzing simulation trajectorials shall allow users to compute various quantities for evaluating and comparing model quality.

We added a new paragraph in the Section: Conclusions and Dicussion of the main text to compare OpenNucleosome with existing software for genome modeling.

“Our software enhances the capabilities of existing genome simulation tools [cite]. Specifically, OpenNucleome aligns with the design principles of Open-MiChroM [cite], prioritizing open-source accessibility while expanding simulation capabilities to the entire nucleus. Similar to software from the Alber lab [cite], OpenNucleome offers highresolution genome organization that faithfully reproduces a diverse range of experimental data. Furthermore, beyond static structures, OpenNucleome facilitates dynamic simulations with explicit representations of various nuclear condensates, akin to the model developed by [citet].”

Comment 3: Lastly, I would appreciate it if the authors could expand their definition of ’standardized practices’.

We apologize for any confusion caused. By ”standardized practices,” we refer to the fact that different groups often employ unique procedures for structural modeling. These procedures differ in the representation of chromosomes, the nucleus environment, and the algorithms for parameter optimization. This absence of a consensus on the optimal practices for genome modeling can be daunting for newcomers to the field.

We have revised the text to the following to avoid confusion:

“Many research groups develop their own independent software, which complicates crossvalidation and hinders the establishment of best practices for genome modeling [3–5].”

Comment 4: On page 7, the authors refer to the SI Section: Components of the whole nucleus model for further details. Could the authors provide more information on the simulated density of nuclear bodies? Is there experimental data available that details the ratio of chromatin to other nuclear components, which was used as a reference in the simulation?

We thank the reviewer for the comment. Imaging studies have provided quantitative measures about the size and number of various nuclear bodies. For example, there are 2 ∼ 5 nucleoli per nucleus, with the typical size RNo ≈ 0.5μm [6–10]. In the review by Spector and Lamond [11], the authors showed that there are 20 ∼ 50 speckles, with the typical size RSp ≈ 0.3μm. We used these numbers to guide our simulation of nuclear bodies. These information was mentioned in the Section: Chromosomes as beads on the string polymers of the supporting information.

The chromatin density is fixed by the average size of chromatin bead and the nucleus size. We chose the size of chromatin based on imaging studies as detailed in the Subsection: Mapping chromatin bead size to real unit of the supporting information. Upon fixing the bead size, the chromatin volume is determined.

Comment 5: In the statement, ’the ideal potential is only applied for beads from the same chromosome to approximate the effect of loop extrusion by Cohesin molecules for chromosome compaction and territory formation,’ it would be helpful if the authors could clarify the scope of this potential. Specifically, the code indicates that the variable ’dend ideal’ is set at 1000, suggesting an interaction along a 100Mb polymer chain at a resolution of 100Kb per bead. Could the authors elaborate on their motivation for the Cohesin complex’s activity having a significant effect over such long distances within the polymer chain?

We thank the reviewer for the insight comment. They are correct that the ideal potential was introduced to capture chromosome folding beyond the interactions between compartments, including loop extrusion. Practically, we parameterized the ideal potential such that the simulated average contact probabilities as a function of sequence separation match the experimental values. The reviewer is correct that beyond a specific value of sequence separation, one would expect the impact of loop extrusion on chromosome folding should be negligible, due to Cohesin dissociation. Correspondingly, the interaction potential should be zero at large sequence separations.

However, it is important to note that the precise separation scale cannot be known a priori. We chose 100Mb as a conservative estimation. However, as we can see from Fig. S7, our parameterization scheme indeed produced interaction parameters are mainly zero at large sequence separations. Interesting, the scale at which the potential approaches 0 (∼ 500KB), indeed agree with the estimated length traveled by Cohesin molecules before dissociation [12].

Comment 6: On pages 8 and 9, the authors discuss the optimization process. However, in reviewing the code and documentation available on the GitHub page, I could not find specific sections related to the optimization procedure described in the paper. In this context, I have a few questions: Could the authors provide more details or direct me to the parts of the documentation and the text/SI that address the optimization procedure used in their study? Additional clarification on the cost/objective function employed during the optimization process would be highly beneficial, as this was not readily apparent in the text.

We thank the reviewer for the comment. We revised the SI to include the definition of the cost function for the Adam optimizer.

“During the optimization process, our aim was to minimize the disparity between experimental findings and simulated data. To achieve this, we defined the cost function as follows:L=∑i(⟨fi⟩−fiexp)2

where the index i iterates over all the constraints defined in Eq. 28.”

The detailed optimization procedure was included in the SI as quoted below

“The details of the algorithm for parameter optimization are as follows

(1) Starting with a set of values for αideal (|i−j|),αcompt (Ti,Tj) and αinter (I,J) we performed 50 independent 3-million-step long MD simulations to obtain an ensemble of nuclear configurations. The 500K steps of each trajectory are discarded

as equilibration. We collected the configurations at every 2000 simulation steps from the rest of the simulation trajectories to compute the ensemble averages defined on the left-hand side of Eq. 13.

(2) Check the convergence of the optimization by calculating the percentage of error

defined as ∑i(⟨fi⟩−fiexp)/∑ifiexp. The summation over i includes all the average contact probabilities defined in Eq. 28.

(3) If the error is less than a tolerance value etol, the optimization has converged, and we stop the simulations. Otherwise, we update the parameters, α, using the Adam optimizer [13]. With the new parameter values, we return to step one and restart the iteration.”

Previously, the optimization code was included as part of the analysis folder. To avoid confusion and improve readability, a separate folder named optimization has been created. This folder provides the Adam optimization of chromosome-chromosome interactions (chr-chr optimization) and chromosome-nuclear landmarks interactions (chr-NL optimization).

Comment 7: What was the motivation for choosing the Adam algorithm for optimization? Adam is designed for training on stochastic objective functions. Could the authors elucidate on the ’stochastic’ aspect of their function to be optimized? Why the Adam algorithm was considered the most appropriate choice for this application?

We thank the reviewer for the comment. As defined in Eq. R1, the cost function measures the difference between the simulated constraints with corresponding experimental values. The estimation of simulation values, by averaging over an ensemble of chromosome configurations, is inherently noisy and stochastic. Exact ensemble averages can only be achieved with unlimited samples obtained from infinite long simulations.

In the past, we have used the Newton’s method for parameterization, and the detailed algorithm can be found in the SI of Ref 14. However, we found that Adam is more efficient as it is a first-order approximation method. The Newton’s method, on the other hand, is second-order approximation method and requires estimation of the Hessian matrix. When the number of constraints is large, as is in our case, the computational cost for estimating the Hessian matrix can be significant. Another advantage of the Adam algorithm lies in its adjustment of the learning rate along the optimization to further speedup convergence.

Comment 8: The authors mention that examples of setting up simulations, parameter optimization, and introducing new features are provided in the GitHub repository. However, I was unable to locate these examples. Could the authors guide me to these specific resources or consider adding them if they are not currently available?

We thank the reviewer for the comment. We have improved the GitHub repository and all the tutorials can be found using the links provided in Response to Comment 2.

Comment 9: Furthermore, the paper states that ’a configuration file that provides the position of individual particles in the PDB file format is needed to initialize the simulations.’ It would be beneficial for new users if the authors could elaborate on how this file is generated. And all other input files in general. Detailing the procedures for a new user to run their system using OpenNucleome would be helpful.

We thank the reviewer for the comment. The procedure for generating initial configurations was explained in the SI Section: Initial configurations for simulations and quoted below.

“We first created a total of 1000 configurations for the genome by sequentially generating the conformation of each one of the 46 chromosomes as follows. For a given chromosome, we start by placing the first bead at the center (origin) of the nucleus. The positions of the following beads, i, were determined from the (i − 1)-th bead as ri=ri−1+0.5v. v is a normalized random vector, and 0.5 was selected as the bond length between neighboring beads. To produce globular chromosome conformations, we rejected vectors, v, that led to bead positions with distance from the center larger than 4σ. Upon creating the conformation of a chromosome i, we shift its center of mass to a value ri com determined as follows. We first compute a mean radial distance, TO2 with the following equation6σ−roiroi−2σ=Dhi−DD−Dlo

where Di is the average value of Lamin B DamID profile for chromosome i. Dhi and Dlo represent the highest and lowest average DamID values of all chromosomes, and 6σ and 2σ represent the upper and lower bound in radial positions for chromosomes. As shown in Fig. S6, the average Lamin B DamID profiles are highly correlated with normalized chromosome radial positions as reported by DNA MERFISH [cite], supporting their use as a proxy for estimating normalized chromosome radial positions. We then select rcomi as a uniformly distributed random variable within the range [roi−2σ,roi+2σ]. Without loss of generality, we randomly chose the directions for shifting all 46 chromosomes.

We further relaxed the 1000 configurations to build more realistic genome structures. Following an energy minimization process, one-million-step molecular dynamics (MD) simulations were performed starting from each configuration. Simulations were performed with the following energy functionURelax =UGenome +UC−LaEV

where UGenome is defined as in Eq. 7. UG-La is the excluded volume potential between chromosomes and lamina, i.e, only the second term in Eq. 24. Parameters in UGenome were from a preliminary optimization. The end configurations of the MD simulations were collected to build the final configuration ensemble (FCE).”

The tutorial for preparing initial configurations can be found at this link.

Comment 10: In the section discussing the correlation between simulated and experimental contact maps, as referenced in Figure 4A and Figure S2, the authors mention a high degree of correlation. Could the authors specify the exact value of this correlation and explain the method used for its computation? Considering that comparing two Hi-C matrices involves a large number of data points, it would be helpful to know if all data points were included in this analysis.

We have updated Fig 4A and S2 to include Pearson correlation coefficients next to the contact maps. The reviewer is correct in that all the non-redundant data points of the contact maps are included in computing the correlation coefficients.

For improved clarity, we added a new section in the supporting information to detail the calculations. The section is titled Computing Pearson correlation coefficients between experimental and simulated contact maps, and the relevant text is quoted below.

“We computed the Pearson correlation coefficients (PCC) between experimental and simulated contact maps in Fig. 4A and Fig. S2 asr=∑i=1n(xi−x¯)(yi−y¯)∑i=1n(xi−x¯)2∑i=1n(yi−y¯)2

xi and yi represent the experimental and simulated contact probabilities, and n is the total number of data points. Only non-redundant data points, i.e., half of the pairwise contacts, are used in the PCC calculation.”

Comment 11: In addition, the author said: ”Moreover, the simulated and experimental average contact probabilities between pairs of chromosomes agree well, and the Pearson correlation coefficient between the two datasets reaches 0.89.” How does this correlation behave when not accounting for polymer compaction or scaling? An analysis presenting the correlation as a function of genomic distance would be interesting.

**Author response image 2. sa4fig2:** Pearson correlation coefficient between experimental and simulated contact probabilities as a function of the sequence separation within specific chromosomes. For each chromosome, we first gathered a set of experimental contacts alongside a matching set of simulated ones for genomic pairs within a particular separation range. The Pearson correlation coefficient at the corresponding sequence separation was then determined using Equation R4. We limited the calculations to half of the chromosome length to ensure the availability of sufficient data.

We thank the reviewer for the comment. The analysis presenting the correlation as a function of genomic distance (sequence separation) for each chromosome is shown in Figure S12 and also included in the SI. While the correlation coefficients decreases at larger separation, the values around 0.5 is quite reasonable and comparable to results obtained using Open-Michrom.

We also computed the correlation of whole genome contact maps after excluding intra-chromosomal contacts. The PCC decreased from 0.89 to 0.4. Again, the correlation coefficient is quite reasonable considering that these contacts are purely predicted by the compartmental interactions and were not directly optimized.

Comment 12: I recommend using the web-server that is familiar to the authors to benchmark the OpenNucleome tool/model: ”3DGenBench: A Web-Server to Benchmark Computational Models for 3D Genomics.” Nucleic Acids Research, vol. 50, no. W1, July 2022, pp. W4-12.

We appreciate the reviewer’s suggestion. Unfortunately, the website is no longer active during the time of the revision. However, as detailed in Response to comment 11, we used the one of the popular metrics to exclude polymer compact effect and evaluate the agreement between simulation and experiments.

Comment 13: Regarding the comparison of simulation results with microscopy data from reference 34. Given their different resolutions and data point/space groupings, how do the authors align these datasets? Could the authors describe how they performed this comparison? How were the radial positions calculated in both the simulations and experiments? Since the data from reference 34 indicates a non-globular shape of the nucleus; how did this factor into the calculation of radial distributions?

We thank the reviewer for the comment and apologize for the confusion. First, the average properties we examined, including radial positions and interchromosomal contacts, were averaged over all genomic loci. Therefore, they are independent of data resolution.

Secondly, instead of calculating the absolute radial positions, which are subject to variations in nucleus shape and size, we defined the normalized radial positions. They measure the ratio between the distance from the nucleus center to the chromosome center and the distance from the nucleus center to the lamina. This definition was frequently used in prior imaging studies to measure chromosome radial positions.

The calculation of the simulated normalized radial positions and the experimental normalized radial positions are discussed in the Section: Computing simulated normalized chromosome radial positions

“For a given chromosome i, we first determined its center of mass position denoted as Ci. Starting from the center of the nucleus, O, we extend the the vector vOC to identify the intersection point with the nuclear lamina as Pi. The normalized chromosome radial position i is then defined as ‖vOCi‖‖vOPi‖, where ||**·||** represents the L2 norm.

and Section: Computing experimental normalized chromosome radial positions.

“We followed the same procedure outlined in Section: Computing simulated normalized chromosome radial positions to compute the experimental values. To determine the center of the nucleus using DNA MERFISH data, we used the algorithm, minimum volume enclosing ellipsoid (MVEE)[15], to fit an ellipsoid for each genome structure. The optimal ellipsoid defined as (x−c)TA(x−c)≡1 is obtained by optimizing min(log⁡(det⁡[A])) subjecting to the constraint that (xi−c)TA(xi−c)≤. xi correspond to the list of chromatin positions determined experimentally.”

Comment 14: In the sentence: ”It is evident that telomeres exhibit anomalous subdiffusive motion.” I recommend mentioning the work ”Di Pierro, Michele, et al., ”Anomalous Diffusion, Spatial Coherence, and Viscoelasticity from the Energy Landscape of Human Chromosomes.” Proceedings of the National Academy of Sciences, vol. 115, no. 30, July 2018, pp. 7753-58.”.

We have revised the sentence to include the citation as follows.

“In line with previous research [cite], telomeres display anomalous subdiffusive motion. When fitted with the equation ⟨r2(Δt)⟩=DαΔtα, these trajectories yield a spectrum of α values, with a peak around 0.59.”

Comment 15: Regarding the observation that ’chromosomes appear arrested and no significant changes in their radial positions are observed over timescales comparable to the cell cycle,’ could the authors provide more details on the calculations or analyses that led to this conclusion? Specifically, information on the equilibration/relaxation time of chromosome territories relative to rearrangements within a cell cycle would be interesting.

Our conclusion here was mostly based on the time trace of normalized radial positions shown in Figure 6A of the main text. Over the timescale of an entire cell cycle (24 hours), the relatively little to no changes in the radial positions supports glassy dynamics of chromosomes. We further determined the mean squared displacement (MSD) for chromosome center of masses. As shown in the left panel of Fig. S12, the MSDs are much smaller than the average size of chromosomes (see Rg values in Fig. 5A), supporting arrested dynamics.

We further computed the auto-correlation function of the normalized chromosome radial position asC(Δt)=∑t=1N(rt−r¯)(rt+Δt−r¯)∑t=1N(rt−r¯)2

where t indexes over the trajectory frames and ¯r is the mean position. As shown in Fig. S12, the positions are not completely decorrelated over 10 hours, again supporting slow dynamics. It would be interesting to examine the relaxation timescale more closely in future studies.

Comment 16: The authors also comment on the SI ”Section: Initial configurations for simulations provides more details on preparing the 1000 initial configurations.” and related to reference 34 mentioning that ”the average Lamin B DamID profiles are highly correlated with chromosome radial positions as reported by DNA MERFISH”. How do the authors account for situations where homologous chromosomes are neighbors or have an interacting interface? Ref 34 indicates that distinguishing between these scenarios can be challenging, potentially leading to ’invalid distributions’ that are filtered out. Clarification on how such cases were handled in the simulations would be helpful.

We would like to first clarify that when comparing with experimental data, we averaged over the homologous chromosomes to obtain haploid data. We added the following text in the manuscript to emphasize this point

“Given that the majority of experimental data were analyzed for the haploid genome, we adopted a similar approach by averaging over paternal and maternal chromosomes to facilitate direct comparison. More details on data analysis can be found in the Supporting Information Section: Details of simulation data analysis.”

Furthermore, we used the processed DNA MERFISH data from the Zhuang lab, which unambiguously assigns a chromosome ID to each data point. Therefore, the issue mentioned by the reviewer is not present in the procssed data. In our simulations, since we keep track of the explicit connection between genomic segments, the trace of individual chromosomes can be determined for any configuration. Therefore, there is no ambiguity in terms of simulation data.

Comment 17: When discussing the interaction with nuclear lamina and nuclear envelop deformation, I suggest mentioning the following studies: The already cited ref 52 and ”Contessoto, Vin´ıcius G., et al. ”Interphase Chromosomes of the Aedes Aegypti Mosquito Are Liquid Crystalline and Can Sense Mechanical Cues.” Nature Communications, vol. 14, no. 1, Jan. 2023, p. 326.”

We updated the text to include the suggested reference.

“Numerous studies have highlighted the remarkable influence of nuclear shape on the positioning of chromosomes and the regulation of gene expression [16, 17].”

Comment 18: The authors state that ’Tutorials in the format of Python Scripts with extensive documentation are provided to facilitate the adoption of the model by the community.’ However, as I mentioned, the documentation appears to be limited, and the available tutorials could benefit from further expansion. I suggest that the authors consider enhancing these resources to better assist users in adopting and understanding the model.

As detailed in the Response to Comment 2, we have updated the GitHub repository to better document the included Jupyter notebooks and tutorials.

Comment 19: In the Methods section, the authors discuss using Langevin dynamics for certain simulations and Brownian dynamics for others. Could the authors provide more detailed reasoning behind the choice of these different dynamics for different aspects of the simulation? Furthermore, it would be insightful to know how the results might vary if only one of these dynamics was utilized throughout the study. Such clarification would help in understanding the implications of these methodological choices on the outcomes of the simulations.

We thank the reviewer for the comment. As detailed in the supporting information Section: Mapping the Reduced Time Unit to Real Time, the Brownian dynamics simulations provide a rigorous mapping to the biological timescale. By choosing a specific value for the nucleoplasmic viscosity, we determined the time unit in simulations as τ = 0.65s. With this time conversion, the simulated diffusion coefficients of telomeres match well with experimental values. Therefore, Brownian dynamics simulations are recommended for computing time dependent quantities and the large damping coefficients mimics the complex nuclear environment well.

On the other hand, the large damping coefficient slows down the configuration relaxation of the system significantly. For computing equilibrium statistical properties, it is useful to use a small coefficient and the Langevin integrator with large time steps to facilitate conformational relaxation.

References

[1] Rao, S. S.; Huntley, M. H.; Durand, N. C.; Stamenova, E. K.; Bochkov, I. D.; Robinson, J. T.; Sanborn, A. L.; Machol, I.; Omer, A. D.; Lander, E. S.; others A 3D map of the human genome at kilobase resolution reveals principles of chromatin looping. Cell 2014, 159, 1665–1680.

[2] Qi, Y.; Zhang, B. Predicting three-dimensional genome organization with chromatin states. PLoS computational biology 2019, 15, e1007024.

[3] Yildirim, A.; Hua, N.; Boninsegna, L.; Zhan, Y.; Polles, G.; Gong, K.; Hao, S.; Li, W.; Zhou, X. J.; Alber, F. Evaluating the role of the nuclear microenvironment in gene function by population-based modeling. Nature Structural & Molecular Biology 2023, 1–14.

[4] Junior, A. B. O.; Contessoto, V. G.; Mello, M. F.; Onuchic, J. N. A scalable computational approach for simulating complexes of multiple chromosomes. Journal of molecular biology 2021, 433, 166700.

[5] Fujishiro, S.; Sasai, M. Generation of dynamic three-dimensional genome structure through phase separation of chromatin. Proceedings of the National Academy of Sciences 2022, 119, e2109838119.

[6] Caragine, C. M.; Haley, S. C.; Zidovska, A. Nucleolar dynamics and interactions with nucleoplasm in living cells. Elife 2019, 8, e47533.

[7] Brangwynne, C. P.; Mitchison, T. J.; Hyman, A. A. Active liquid-like behavior of nucleoli determines their size and shape in *Xenopus laevis* oocytes. Proceedings of the National Academy of Sciences 2011, 108, 4334–4339.

[8] Farley, K. I.; Surovtseva, Y.; Merkel, J.; Baserga, S. J. Determinants of mammalian nucleolar architecture. Chromosoma 2015, 124, 323–331.

[9] Qi, Y.; Zhang, B. Chromatin network retards nucleoli coalescence. Nature Communications 2021, 12, 6824.

[10] Caragine, C. M.; Haley, S. C.; Zidovska, A. Surface fluctuations and coalescence of nucleolar droplets in the human cell nucleus. Physical review letters 2018, 121, 148101.

[11] Spector, D. L.; Lamond, A. I. Nuclear speckles. Cold Spring Harbor perspectives in biology 2011, 3, a000646.

[12] Banigan, E. J.; Mirny, L. A. Loop extrusion: theory meets single-molecule experiments. Current opinion in cell biology 2020, 64, 124–138.

[13] Kingma, D. P.; Ba, J. Adam: A method for stochastic optimization. arXiv preprint arXiv:1412.6980 2014,

[14] Zhang, B.; Wolynes, P. G. Topology, structures, and energy landscapes of human chromosomes. Proceedings of the National Academy of Sciences 2015, 112, 6062–6067.

[15] Moshtagh, N.; others Minimum volume enclosing ellipsoid. Convex optimization 2005, 111, 1–9.

[16] Brahmachari, S.; Contessoto, V. G.; Di Pierro, M.; Onuchic, J. N. Shaping the genome via lengthwise compaction, phase separation, and lamina adhesion. Nucleic Acids Res. 2022, 50, 1–14.

[17] Contessoto, V. G.; Dudchenko, O.; Aiden, E. L.; Wolynes, P. G.; Onuchic, J. N.; Di Pierro, M. Interphase chromosomes of the Aedes aegypti mosquito are liquid crystalline and can sense mechanical cues. Nature Communications 2023, 14, 326.